# Source attribution of cloud condensation nuclei and their impact on stratocumulus clouds and radiation in the south-eastern Atlantic

Haochi Che[1, a], Philip Stier[1], Duncan Watson-Parris[1], Hamish Gordon[2, b], Lucia Deaconu[1, c]

[1]Atmospheric, Oceanic and Planetary Physics, Department of Physics, University of Oxford, Oxford, OX1 3PU, UK

[2]School of Earth and Environment, University of Leeds, LS2 9JT, UK

[a]now at: Department of Geophysics, Tel-Aviv University, 69978, Israel

[b]now at: Engineering Research Accelerator, Carnegie Mellon University, Pittsburgh, PA 15217, United States

[c]now at: Faculty of Environmental Science and Engineering, Babeș-Bolyai University, Cluj-Napoca, 400294, Romania

*Correspondence to*: Haochi Che (haochiche@tauex.tau.ac.il)

**Abstract** The semi-permanent stratocumulus clouds over the South-eastern Atlantic Ocean (SEA) can act as an "air conditioner" to the regional and global climate system. The interaction of aerosols and clouds becomes important in this region, and can lead to negative radiative effects, partially offsetting the positive radiative forcing of greenhouse gases. A key pathway by which aerosols affect cloud properties is by acting as cloud condensation nuclei (CCN). In this paper, we use the United Kingdom Earth System Model to investigate the sources of CCN (from emission and atmospheric processes) in the SEA, and the response of cloud droplet number concentration (CDNC), cloud liquid water path (LWP), and radiative forcing to those sources during 2016 and 2017. Overall, free and upper troposphere nucleated aerosols are the dominant source of boundary layer $CCN_{0.2\%}$, contributing an annual average of ~ 41 % as they subside and entrain into the marine boundary layer, which is consistent with observations highlighting the important role of nucleation for boundary layer CCN. In terms of emission sources, anthropogenic emissions (from energy, industry, agriculture, etc.) contribute the most to the annual average $CCN_{0.2\%}$ in the marine boundary layer (~ 26 %), followed by biomass burning (BB, ~ 17 %). In the cloud layer, BB contributes about 34 % of annual $CCN_{0.2\%}$, midway between the contributions from aerosol nucleation (36%) and anthropogenic sources (31%). The contribution of aerosols from different sources to CDNC is consistent with their contribution to $CCN_{0.2\%}$ within the marine boundary layer, with free and upper troposphere aerosol nucleation being the most important source of CDNC overall. In terms of emission sources, anthropogenic sources are also the largest contributors to the annual average of CDNC, closely followed by BB. However, during the BB season, BB and free and upper troposphere aerosol nucleation are equally the most important sources of CDNC. The contribution of BB to CDNC is more significant than its increase to $CCN_{0.2\%}$, mainly because BB aerosols are mostly located directly above the inversion layer in the model, thus can increase in-cloud CDNC by enhancing the supersaturation through the dynamical feedback due to shortwave absorption. For an aerosol source that shows an increase in CDNC, it also shows an increase in LWP resulting from a reduction in autoconversion. Due to the absorption effect, BB aerosol can enhance existing temperature inversions and reduce the entrainment of sub-saturated air, leading to a further

increase in LWP. As a result, the contribution of BB to LWP is second only to aerosol nucleation on annual averages. These findings demonstrate that BB is not the dominant source of CCN within the marine boundary layer from an emission source perspective. However, as most BB aerosols are located directly above the inversion layer, their effect on clouds increases due to its absorption effect (about the same as anthropogenic sources for CDNC and more than anthropogenic sources for LWP),

highlighting the crucial role of its radiative effect on clouds. The results on the radiative effects of aerosols show that BB aerosol exhibits an overall positive $RF_{ari}$ (radiative forcing associated with aerosol-radiation interaction), but its net effective radiative forcing remains negative due to its effect on clouds (mainly by absorbing effect). By quantifying aerosol and cloud properties affected by different sources, this paper provides a framework for understanding the effects of aerosol sources on the marine stratocumulus clouds and radiation in the SEA.

**1 Introduction**

Marine stratocumulus clouds cover approximately one-quarter of the ocean surface in the annual mean (Hahn and Warren, 2007), resulting in a strong negative net radiative effect that significantly affects climate, therefore referred to as "air conditioners" in the climate system (Stephens and Slingo, 1992). Stratocumulus clouds in the South-eastern Atlantic Ocean (SEA) are one of the most extensive stratocumulus cloud decks on the planet (Wood, 2012) and are semi-permanently present

off the coast of Africa. The interaction of aerosol-cloud becomes extremely important in this region, as a moderate change in the cloud coverage or liquid water path induced by aerosol could compensate for the radiative forcing of greenhouse gases and significantly affect the regional or global climate (Wood, 2012). A key pathway of aerosol effects on cloud properties is by acting as cloud condensation nuclei (CCN). The increased CCN from different sources can alter the liquid water path (LWP) by affecting the cloud state (Berner et al., 2015) and cloud lifetime (Ackerman et al., 2004). At a fixed LWP, the increased

CCN due to emission perturbations leads to the increase of cloud droplet concentrations of smaller radii and subsequently to an increase of cloud albedo, commonly referred to as the radiative forcing associated with aerosol-cloud interaction, i.e. $RF_{aci}$ (Twomey, 1974). The increased CCN can also trigger rapid adjustments, affecting the cloud lifetime and precipitation (Albrecht, 1989). Under the combined effects of these two ($RF_{aci}$ and rapid adjustments), referred to as $ERF_{aci}$, aerosol-cloud interactions represent one of the largest sources of uncertainties in future climate projections (Boucher et al., 2013).

In the SEA, the persistent stratocumulus cloud deck encounters particles from various sources. Among them, biomass burning (BB) aerosol, advected from continental Africa, where one-third of the global BB emissions are produced from July to October (Roberts et al., 2009; van der Werf et al., 2010), plays a unique role in modulating the cloud properties due to its shortwave absorption ability as well as acting as CCN. Previous studies suggest that as the BB aerosols are mainly located above and

near the inversion layer, when above the inversion layer, the main role of their radiative effect in the SEA is to strengthen the capping inversion and reduce dry air entrainment from cloud tops, thereby increasing the LWP and low-level cloud fraction,

resulting in a significant impact on the radiation balance (Wilcox, 2010; Gordon et al., 2018; Deaconu et al., 2019; Mallet et al., 2020; Herbert et al., 2020; Chaboureau et al., 2022). When BB aerosols are located in the marine boundary layer, their radiative effect can enhance the decoupled boundary layer and result in a reduction in cloud cover and LWP, shifting the stratocumulus-to-cumulus transition to the upwind area (Zhang and Zuidema, 2019; Ajoku et al., 2021). CCN from BB can

play an important role in affecting stratocumulus cloud droplet concentration and the radiative forcing (Lu et al., 2018). However, there exists no consensus on the importance of BB aerosol in acting as CCN (Che et al., 2021; Gordon et al., 2018; Lu et al., 2018; Mallet et al., 2020) in the SEA, mainly because the fraction of BB aerosol entering the marine boundary layer remains uncertain. Sea-salt aerosol is one of the largest contributors to global primary aerosols in terms of mass concentration (Seinfeld and Pandis, 1998). Although its particles can be easily activated due to their high hygroscopicity, the contribution of

sea-salt aerosol to the marine CCN population and its relative importance in indirect effects over the ocean is uncertain (Tsigaridis et al., 2013). Some studies report sea-salt aerosol as the primary source of CCN over the ocean (e.g. Pierce and Adams, 2006), while other studies found sea-salt only contributes a small fraction of marine CCN (Quinn et al., 2017). Besides sea salt, marine emissions are also the primary source of dimethyl sulfide (DMS), which produces the largest fraction of natural sulfur species in the atmosphere via oxidation (Andreae, 1990). The oxidation products of DMS (methanesulfonic acid and

$H_2SO_4$) can form new particles via multiple aerosol nucleation processes (i.e., binary, ternary, and ion-induced) or condense onto existing particles and eventually form CCN (e.g., Lee et al., 2003). CCN from DMS is crucial for marine boundary layer clouds and is often found to have a key role in the clean and low wind marine environment (Sanchez et al., 2018), resulting in profound climate implications (Charlson et al., 1987; Thomas et al., 2010). With the development of the African economy, anthropogenic emissions from energy, industrial, agriculture etc., are expected to increase significantly and could have a

similar magnitude to that of African biomass burning around 2030 (Liousse et al., 2014). Many studies have concluded that anthropogenic aerosols are generally hygroscopic and responsible for the increase of global CCN (Che et al., 2017; Rose et al., 2011; Schmale et al., 2018; Yu et al., 2013). In the SEA, most anthropogenic and BB aerosols are advected from continental Africa and can be activated to cloud droplets by entraining clouds. However, even under a similar amount of advected concentration, these two types of aerosols may contribute differently to the number of activated droplets in the stratocumulus

clouds layer due to their different abilities to affect the atmosphere temperature profile. Another potentially important source of CCN is dust. Although insoluble, wettable dust particles with large diameters can act as CCN, while small dust particles can accumulate soluble materials through internal mixing during transportation, and dramatically increase their ability to activation (Bègue et al., 2015; Dusek et al., 2006; Gibson et al., 2007; Hatch et al., 2008).

Apart from primary emissions, a large fraction of atmospheric aerosols, known as secondary aerosols, are formed from atmospheric processes (oxidation of gaseous precursors, i.e., aerosol nucleation) and can serve as CCN after the subsequent growth of nucleated clusters to sufficiently larger sizes (Kerminen et al., 2012; Merikanto et al., 2009a). Studies have found that the nucleation of aerosols in the boundary layer and free troposphere is the dominant source of particle number in the atmosphere (Kulmala et al., 2004; Kulmala and Kerminen, 2008). However, the contribution of aerosol nucleation to CCN is

not consistent, as some studies found that less than 10% of nucleated particles can grow to diameters of 100 nm in general (Kuang et al., 2009; Westervelt et al., 2013), implying the potentially limited role of aerosol nucleation in providing CCN. While several studies have shown that small particles generated by aerosol nucleation in the free troposphere can grow with subsidence and contribute more than half of CCN in the global marine boundary layer (Clarke et al., 2013; Merikanto et al., 2009a; Williamson et al., 2019; Clarke and Kapustin, 2002). Therefore, the role of aerosol nucleation in affecting stratocumulus cloud deck in SEA remains uncertain, hindering our understanding of the aerosol-cloud interactions in this region.

Two aircraft observation campaigns were performed during the BB season in the SEA to enable an intensive study of the aerosols and clouds interactions in this region. Those two campaigns flew different areas in the SEA, where the NASA ORACLES (ObseRvations of Aerosols above CLouds and their intEractionS) were launched from Walvis Bay (Namibia) in 2016 and from Sao Tomé in 2017 (Redemann et al., 2021); CLARIFY (CLoud–Aerosol–Radiation Interaction and Forcing: Year 2017) flew around Ascension Island and can provide information on aerosol-cloud-radiation interactions in the region where stratocumulus to cumulus transition occurs (Haywood et al., 2021). In addition, a ground-based in-situ field measurement campaign (LASIC, Layered Atlantic Smoke Interactions with Clouds) was carried out on Ascension Island, which provided 18 months of observations for aerosols and clouds within the marine boundary layer from June 2016 to October 2017 (Zuidema et al., 2018b). The main focus of these campaigns is BB aerosols and their effects on clouds. Here we use United Kingdom Earth System Model, UKESM1(Sellar et al., 2019) to investigate the source attribution of CCN in the SEA, contributing to the understanding of the main sources of CCN in this region. The subsequent cloud droplet concentration, cloud liquid water and radiative forcing associated with different sources are also investigated. The model has previously been evaluated with data from ORACLES and CLARIFY observations and the results show that it provides a good simulation of the spatial and vertical distribution of aerosols (Che et al., 2021). The article is structured as follows. Section 2 presents the method, including model configuration and evaluation; the results are listed in section 3. In section 3.1, we investigate the vertical distribution of CCN contributed by emissions and atmospheric processes, as well as the mean concentrations of CCN in different layers. Section 3.2 examines the cloud adjustments by aerosols from different sources, including CDNC and LWP. Section 3.3 shows radiative forcings associated with those aerosols. Section 4 contains conclusions and discussions.

## 2 Method

### 2.1 Model configuration

The first version of the United Kingdom Earth System Model, UKESM1(Sellar et al., 2019), has been jointly developed by the UK's Met Office and Natural Environment Research Council (NERC). The core of UKESM1 is based on the Hadley Centre Global Environmental Model version 3 (HadGEM3) Global Coupled (GC) climate configuration of the Unified Model (UM) (Hewitt et al., 2011). The atmospheric part of the model is configured as Global Atmosphere 7.1 (GA7.1) (Walters et

al., 2019). Aerosol and its interactions with clouds are represented by the UK Chemistry and Aerosol model (UKCA) (Mulcahy et al., 2020; O'Connor et al., 2014). Differing from the standard configuration of representing the dust size distribution as six bins (Woodward, 2001), our configuration uses seven interactive log-normal aerosol modes in the microphysics scheme GLOMAP (Mann et al., 2010), comprising sulfate, sea salt, black carbon, organic carbon and dust, allowing for condensation and coagulation on or with dust. With this setting, we can set the hygroscopicity of different aerosol species with a single parameter $\kappa$. The bulk properties (cloud fraction, cloud liquid water content, etc.) of large-scale clouds are parameterized using the prognostic cloud fraction and prognostic condensate (PC2) scheme (Wilson et al., 2008a, b) with modifications described in Morcrette (2012). Cloud droplet is derived using the activation scheme of Abdul-Razzak and Ghan (2000). The activated CCN can be expressed as a function of aerosol properties (size, number, and composition) and thermodynamic properties (e.g., updraught velocity, temperature, pressure, and specific humidity), where thermodynamic properties are used to determine the local supersaturation, and aerosol properties are used to calculate activated CCN. When the local supersaturation has been determined, activated CCN is calculated with the $\kappa$-Kohler scheme, which uses a parameter $\kappa$ to represent the hygroscopicity of aerosols. The $\kappa$ value is set as 0.6, 0.2, and 1.2 for sulfate, organic, and sea-salt, respectively, and 0 for black carbon and dust (Engelhart et al., 2012; Petters and Kreidenweis, 2007). The internal volume mixing rule (Petters and Kreidenweis, 2007) is used to calculate the mean hygroscopicity of each mode. Therefore, a higher fraction of less hygroscopic components (e.g. organic and black carbon) can reduce the overall $\kappa$. However, the overall κ may be underestimated when BC has a thicker coating. This was illustrated by Kacarab et al. (2020), who found a high averaged κ of ~ 0.4 from eight ORACLES 2017 aircraft observations. However, Zhang et al. (2022) found an averaged κ of ~ 0.24 in the marine boundary layer from ORACLES 2018 observations, which is consistent with our assumption that BB reduces the overall κ. Cloud droplet concentrations at the cloud base are replicated vertically throughout contiguous columns of the cloud. After running the cloud activation scheme, the CDNC is then passed to the radiation and microphysics schemes. The Coupled Model Intercomparison Project Phase 6 (CMIP6) emission data in 2014 is used (Eyring et al., 2016; Gidden et al., 2019). Due to the high interannual variability of BB emission, the global fire assimilation system (GFAS) version 1 data based on satellite fire monitoring is employed with a scaling factor of 2 (Johnson et al., 2016).

The GFAS biomass burning and CMIP6 2014 emissions are used as the baseline simulation. To facilitate source attribution, four additional runs are made with BB, dust, sea-salt, and DMS emission turned off, and one simulation with pre-industrial CMIP6 emissions. The effect of these sources on aerosols, clouds and radiation can then be derived from the difference between the baseline simulation, and the individual runs with emissions turned off. The different aerosol sources (anthropogenic sources, biomass burning, etc.) are defined from the perspective of CMIP6. Note that although black carbon (BC) and organic carbon (OC) are the main components of BB emissions, these two types of aerosols are also present in anthropogenic emissions. However, the 'anthropogenic' emissions defined here do not include BB aerosols, although BB in southern Africa is associated with human activities (Roberts et al., 2009). In our model setup, BC and OC from our 'anthropogenic' emissions are mainly from fossil fuels and biofuels, and their emission sectors are energy, industrial, shipping, transportation, solvents, waste,

agriculture, and residential. In comparison, BC and OC from BB are mainly emitted from the burning of agricultural land, peat, savanna, forest, and deforestation. It also should be noted that changes in emissions of aerosols or their precursors can affect the chemical and microphysical ageing capacity of the atmosphere, resulting in a nonlinear response of aerosol populations. Stier et al. (2006) investigated the nonlinear responses and found they are generally not dominant and manifested in alterations

of the aerosol lifetimes. In addition to emission sources, three additional runs are performed without SOA (secondary organic aerosol) formation, boundary layer aerosol nucleation, and total aerosol nucleation, to allow attribution to atmospheric processes. The boundary aerosol nucleation scheme is based on the organic-mediated nucleation parameterization of Metzger et al., (2010), determined by the concentrations of sulfuric acid and SOA, and limited to the boundary layer. The total aerosol nucleation includes the boundary layer nucleation and homogeneous binary aerosol nucleation of sulphuric acid and water,

which is applicable to both tropospheric and stratospheric conditions, as described in Vehkamäki et al. (2002). Precursors for aerosol nucleation include $H_2SO_4$, which is contained in both natural and anthropogenic emissions. Therefore, aerosol nucleation can also be affected by these emissions, particularly in areas with strong anthropogenic emissions (Saha et al., 2018). However, the exact extent of the impact of these emissions on nucleation remains unresolved in this work and requires future analysis. The gas-phase oxidations of Monoterpene by OH, $O_3$, and $NO_3$ yield SOA at a fixed rate of 0.26 (unitless, denotes

26% production percentage), which is scaled by 2 from the original value based on alpha-pinene (Spracklen et al., 2006) to compensate for the missing SOA from other sources. The contributions of these atmospheric processes are also derived from differences between the baseline simulation and individual runs. The resolution of our simulations is N96, i.e., 1.875º × 1.25º, with 85 vertical levels. Sea surface temperatures and sea ice are prescribed with daily reanalysis data (Reynolds et al., 2007). In all runs, horizontal wind fields above 1500 m are nudged every 6 h with ERA-Interim data (Telford et al., 2008), while the

temperature is free-run to allow fast adjustments, following the recommendations of Zhang et al. (2014).

## 2.2 Investigated area and time period

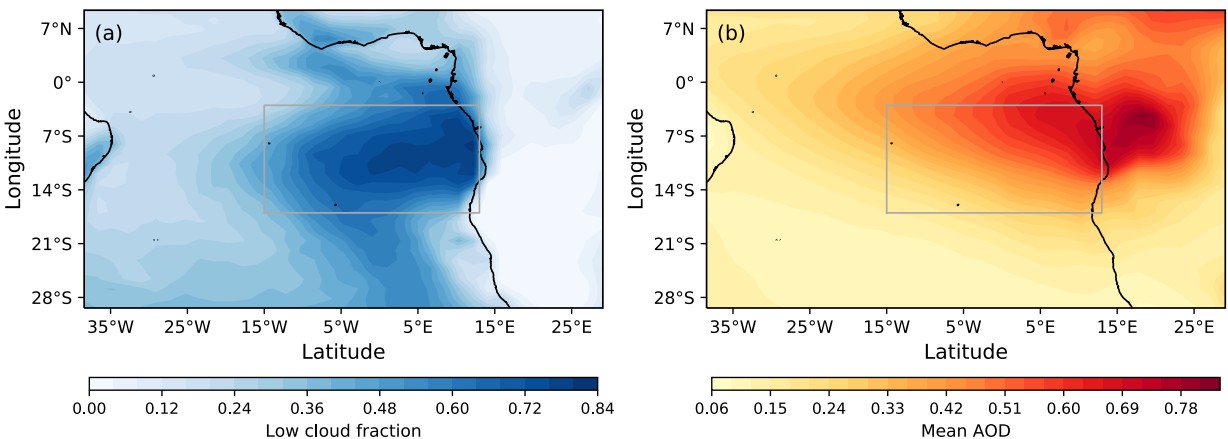

Figure 1. UKESM1 simulated mean (a) low-level cloud fraction and (b) aerosol optical depth from July to September 2016 and 2017. The domain, ranging from 30° S to 10° N and from 40° W to 30° E, is the area this paper interested in. The grey box (cloud box) in the map represents the area where the average low cloud fraction is above 0.6.

The model runs from 2016 to 2017 to overlap with the ORACLES, CLARIFY and LASIC observation campaigns conducted in the SEA. July, August and September of the two years are selected to represent the BB season, as the highest mean AOD associated with BB are found in these months. The low-level cloud fraction during the BB season is higher than the annual average, indicating the co-existence of the intensive stratocumulus cloud deck and BB plume in the SEA. Fig. 1 shows the mean low cloud fraction and AOD during BB season. In addition to the illustrated domain representing the SEA region in the

figure, a small area is identified as the cloud box region, with the fraction of low-level clouds at 0.6 for the BB season mean and 0.5 for the two-year average. The location and size of the cloud box region are different from the one identified by Klein and Hartmann (1993), as we encompass stratocumulus and cumulus clouds. Despite the border of our defined area, the annual mean of low cloud fraction is 0.5 in the cloud box, indicating the semi-permanent feature of low clouds in this region. The mean AOD in the cloud box region is 0.43 and 0.26 for the BB season and the two-year averages separately. BB aerosol

contributes around 76 % of total AOD in the cloud box during BB season, and can result in a clearly elevated CDNC in the SEA from satellite observations (Redemann et al., 2021), implying the potentially dominant role of BB aerosol in affecting CCN and cloud that motivated the ORACLES, CLARIFY and LASIC campaigns. However, as most of the BB aerosol is above the stratocumulus cloud deck (Fig. 2), combined with a large fraction of low-hygroscopicity particles such as BC and OC, the fraction of BB aerosol to activate as cloud droplets is uncertain.

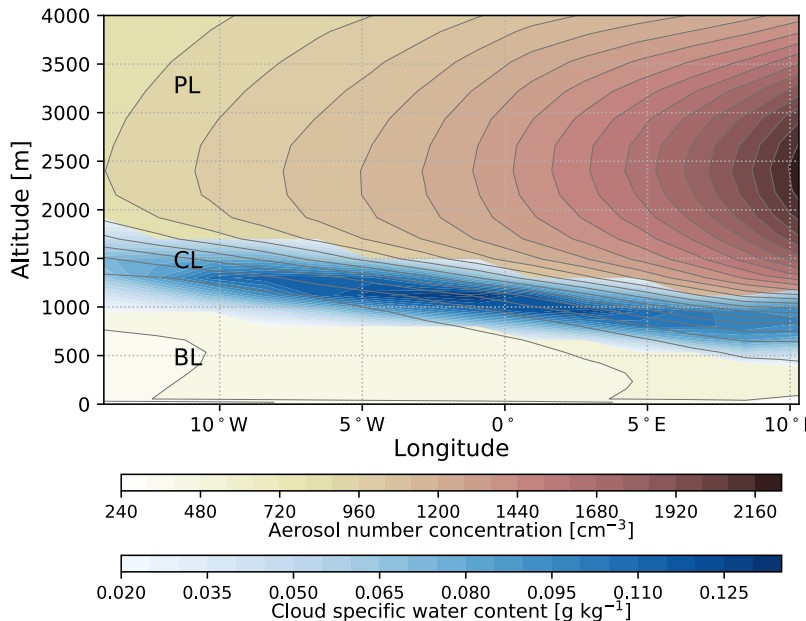

Figure 2. UKESM1 simulated mean vertical profiles of cloud specific water content (g/kg) and aerosol number concentration (cm$^{-3}$) at the standard temperature and pressure (STP) in the cloud box region during BB season. The BL, CL, and PL represent the boundary layer, cloud layer, and plume layer, respectively. The cloud layer is identified as the layer with specific liquid water content > 0.02 g/kg.

The mean vertical profiles of cloud liquid water content and aerosol number concentration in the cloud box during BB season are illustrated in Fig. 2. Three layers are defined to investigate source attributed CCN in different areas. The cloud layer (CL), where liquid water content is above 0.02 g/kg, represents the semi-permanent stratocumulus cloud deck. We define the area below the cloud layer as the boundary layer (BL), and above the plume layer (PL). As shown in Fig. 2, most aerosols emanating from the continent are located in the plume layer, with a maximum concentration at the height of around 2500m. The boundary layer has the lowest aerosol concentration. This may be because only a small proportion of aerosols can enter the cloud layer from the top, and the fraction of aerosols that could enter the boundary layer is further reduced by the cloud wet scavenging process (Textor et al., 2006). Therefore, the boundary layer is relatively clean, with the aerosol concentration around a few hundred per cubic centimetre. However, as the boundary layer is close to the sea surface, it contains a higher proportion of more hygroscopic sea-salt aerosols. The annual mean vertical profiles of liquid water content and aerosol number concentration in the cloud box have a similar pattern, with a lower concentration of aerosols and cloud liquid water (Fig. S1 in supplement).

**2.3 Model evaluation**

The model has been evaluated with the ORACLES (2016, 2017) and CLARIFY measurements by examining the collocated aerosol extinction in our previous paper. The result shows the model can generally capture the spatial and vertical distributions of BB plume (Che et al., 2021). However, as these aircraft observations are mainly located in the free troposphere, we further
5    evaluated modelled CCN within the marine boundary layer using LASIC observations.

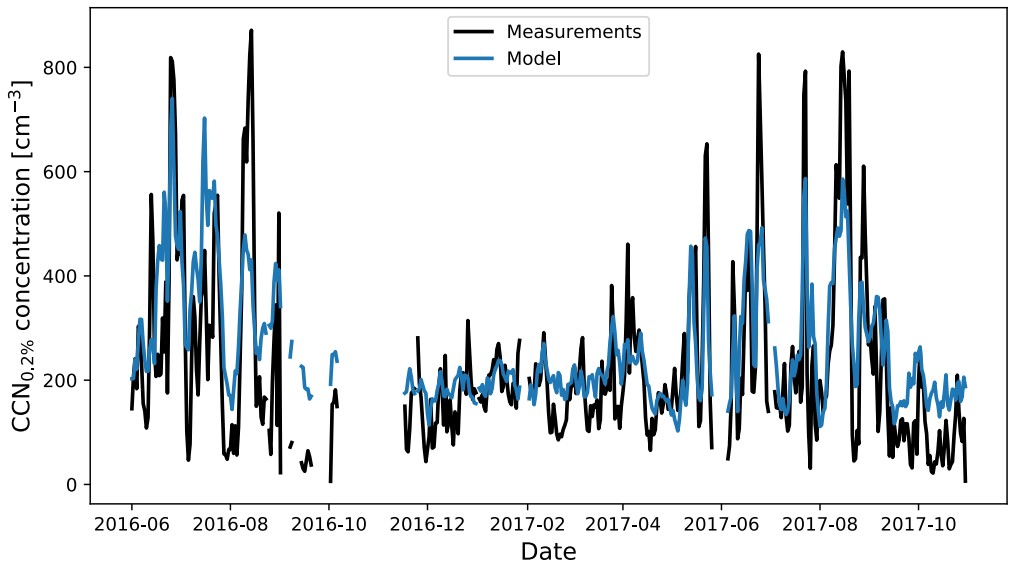

Figure 3. Comparison of modelled and observed daily mean $CCN_{0.2\%}$ (CCN at 0.2 % supersaturation) concentrations. The measured $CCN_{0.2\%}$ is from the LASIC campaign. The modelled $CCN_{0.2\%}$ is from the baseline simulation and interpolated to the LASIC coordinates.

The LASIC campaign was carried out on the Atmospheric Radiation Measurement (ARM) Mobile Facility 1 site at Ascension Island, located at a latitude of -7.97°, longitude of-14.35° and altitude of 340.7664 m. The LASIC CCN was measured by a cloud condensation nuclei counter (CCNC-200), which provides the CCN concentration at fixed supersaturations (Roberts and Nenes, 2005; Atmospheric Radiation Measurement (ARM) user facility, 2016). A more detailed description of the sampling
15    location and instruments can be found in the campaign report (Zuidema et al., 2018a). The modelled CCN concentration at 0.2% supersaturation ($CCN_{0.2\%}$) from the baseline simulation is collocated with observations. Due to the temporal resolution of the model output, we compared the daily averages as illustrated in Fig. 3.

As evident in Fig. 3, the modelled $CCN_{0.2\%}$ is in good agreement with the observation, and can capture the daily variation of
20    $CCN_{0.2\%}$ during the BB season. The campaign averaged $CCN_{0.2\%}$ is 225 cm$^{-3}$, and the modelled corresponding means of 239

cm⁻³, with the mean relative error of the modelled CCN$_{0.2\%}$ around 6.3%. However, the observed CCN peaks during the BB season are higher than simulations, indicating that the model is still inadequate for capturing those peak values. One possible reason is that when BC particles have a thick coating, the calculated overall κ may be underestimated by the volume mixing rule, which may further underestimate the CCN concentration associated with BB (Kacarab et al., 2020). In addition, uncertainties in the BB emissions, including the magnitude, size and, height of fires, can lead to incorrect estimates of BB aerosol peak concentrations, which can lead to such underestimations of CCN. Given that we mainly investigate the annual mean CCN in this study, the relatively small error and the well-matched temporal variability with observation suggest that the model is fairly reasonably in reproducing the CCN in the marine boundary layer in the SEA. Therefore, this result provides confidence in this study.

# 3 Results

## 3.1 Cloud condensation nuclei concentration

### 3.1.1 Vertical distribution of CCN

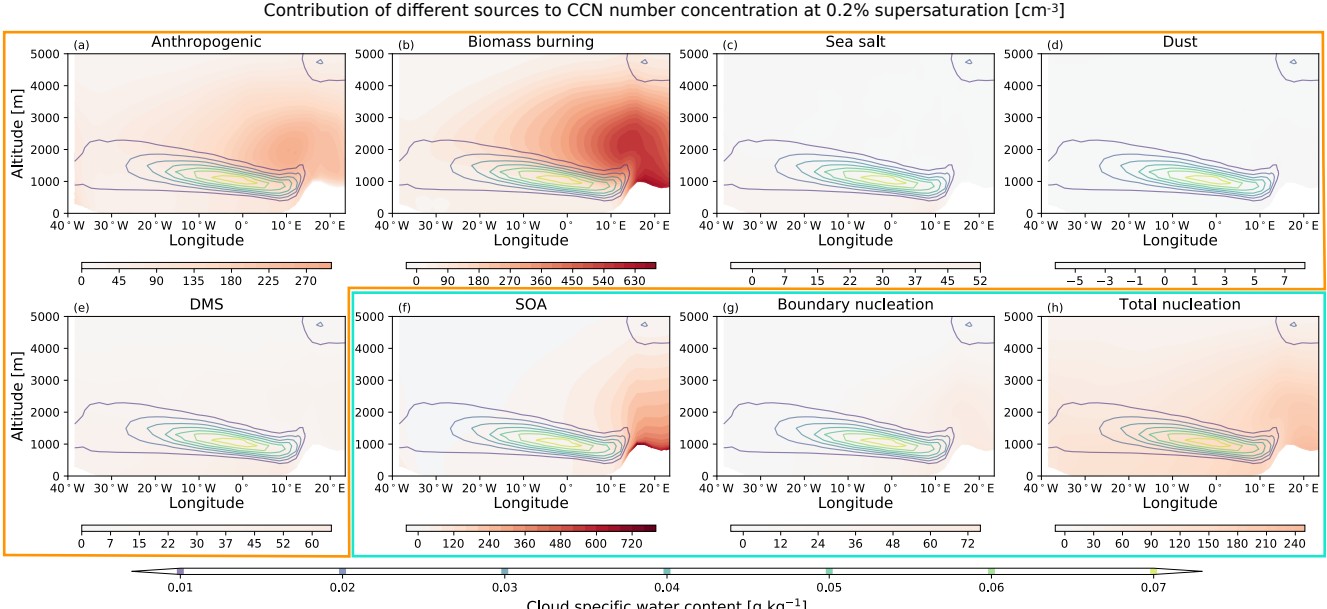

Figure 4. UKESM1 simulated annual mean vertical profiles of CCN concentration at 0.2% supersaturation (CCN$_{0.2\%}$) from different sources (at the standard temperature and pressure STP). Profiles are averaged along the latitudes of the cloud box. The contributions of different sources to CCN$_{0.2\%}$ are listed in (a) to (h), where the contribution of emissions is shown in the yellow frame, and the contribution of atmospheric processes is shown in the light blue frame. Note boundary layer aerosol

nucleation is based on organic-mediated aerosol nucleation and is limited to the boundary layer. Total aerosol nucleation includes boundary layer nucleation and homogeneous binary aerosol nucleation in the free troposphere and stratosphere. The contour lines in each subplot are the cloud specific water content from the baseline simulation at the same temporal and spatial average. The same colourmap scale is used in each subplot to facilitate comparison, but the range differs for each plot, corresponding to the maximum and minimum of $CCN_{0.2\%}$.

The annual mean profiles of $CCN_{0.2\%}$ (CCN at 0.2% supersaturation) concentration from different sources are illustrated in Fig. 4. Overall, BB is the dominant source of $CCN_{0.2\%}$, although its contribution is mainly distributed above the cloud layer. This is because BB aerosol is emitted from the continent and is mainly located in the free troposphere in its westward transport. Anthropogenic aerosol, also originated from the land, is the second-largest source of $CCN_{0.2\%}$ from emissions in the SEA, while its $CCN_{0.2\%}$ concentration is around one-third of that associated with BB above the cloud. However, when in the cloud layer, these two sources are almost equally important, with the $CCN_{0.2\%}$ from BB being only slightly higher. This may be partly because the shortwave absorption capability of BB aerosol has inhibited the cloud top entrainment when the BB aerosol is above the clouds (Johnson et al., 2004; Sakaeda et al., 2011; Wilcox, 2010), resulting in fewer BB aerosols being able to enter the clouds. Another $CCN_{0.2\%}$ source linked strongly with the land is SOA, as monoterpene, the precursor of the SOA in our model, is mostly from plants (Mentel et al., 2009). Marine emissions make a small contribution to monoterpene (Yassaa et al., 2008), contributing to SOA concentrations in the marine boundary layer.

DMS and sea-salt attributed $CCN_{0.2\%}$ have low concentrations and are mainly distributed in the marine boundary layer, as they are both emitted from the ocean surface. Although aerosols from these two sources have high hygroscopicity, their low number concentration limited their CCN number. Dust merely has an impact on the $CCN_{0.2\%}$, partly due to the hydrophobic characteristics of its particles as represented in the model, and partly because of the low concentration of dust in this region. The reduction (negative) in CCN due to dust may be due to the increase in sulphuric acid condensation sink, thereby reducing the CCN from aerosol nucleation. For atmospheric processes, both the total and boundary layer aerosol nucleation lead to an increase in $CCN_{0.2\%}$ concentrations, indicating the contribution of aerosol nucleation to CCN. However, their contribution to $CCN_{0.2\%}$ is lower than that of BB and anthropogenic emissions above clouds. Total aerosol nucleation contributes more to $CCN_{0.2\%}$ compared to boundary layer nucleation, indicating a contribution from the free and upper troposphere. The $CCN_{0.2\%}$ mean concentration profiles during the BB season show a similar pattern to the annual means (Fig. S2), while the $CCN_{0.2\%}$ concentration associated with BB, anthropogenic emission and SOA is ~ 2.3, 1.8, and 1.5 times its annual means, respectively, indicating the significant contribution of BB to CCN.

### 3.1.2 Mean concentration of CCN at different layers

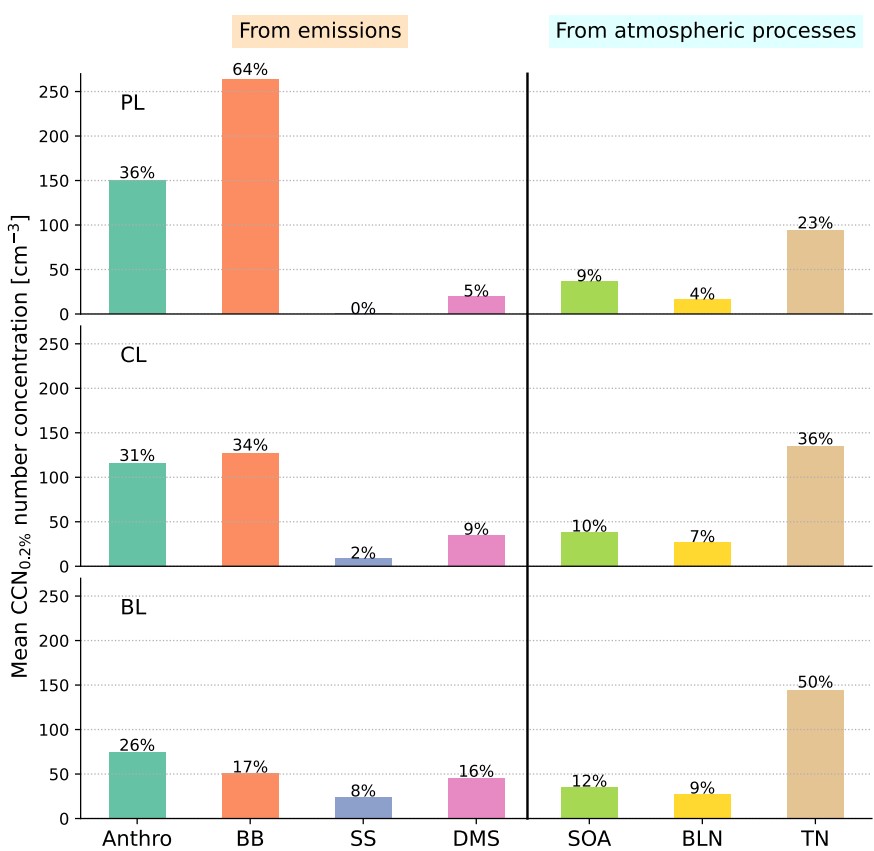

Figure 5. UKESM1 simulated annual mean CCN concentration at 0.2% supersaturation ($CCN_{0.2\%}$) in the cloud box region from different sources and in different layers. The upper, middle and lower panels represent $CCN_{0.2\%}$ attribution in the plume layer (PL), cloud layer (CL), and the marine boundary layer (BL), respectively. The left part of the black vertical line indicates the contribution to $CCN_{0.2\%}$ from the emission sources, which are anthropogenic (Anthro), BB, sea salt (SS), and DMS. The right part of the black vertical line indicates the contribution to $CCN_{0.2\%}$ from atmospheric processes, which are SOA, boundary layer nucleation (BLN) and total aerosol nucleation (TN). BLN is based on the organic-mediated aerosol nucleation and is limited to the boundary layer. TN includes BLN and homogeneous binary aerosol nucleation in the free troposphere and stratospheric. Using the simulation of the present day as the baseline (annual mean $CCN_{0.2\%}$ around 290 cm$^{-3}$), the contribution of each source to $CCN_{0.2\%}$ is marked at the top of the corresponding bar in percentage.

The source attribution of CCN in the cloud box region is investigated in this section. The location and definition of the cloud box region can be found in the methods. Here we also focus on the CCN concentration at 0.2% supersaturation, as the maximum

supersaturation in the area is usually less than 0.2% (Che et al., 2021). The annual mean concentrations of $CCN_{0.2\%}$ in the cloud box at different layers contributed by different sources are illustrated in Fig. 5. The contribution to $CCN_{0.2\%}$ from different emissions is shown on the left of the black vertical line, while the contribution to $CCN_{0.2\%}$ from atmospheric processes is on the right. Overall, the most important source of $CCN_{0.2\%}$ in the marine boundary layer (BL) is total aerosol nucleation.

This is evident in both the two-year and the BB season averages (Fig. S3). As most of the aerosol nucleation occurs in the free and upper troposphere, this result suggests that the subsidence and growth of free and upper tropospheric nucleated aerosols contribute significantly to the CCN in the marine boundary layer. The boundary layer aerosol nucleation contributes about 9 % of the $CCN_{0.2\%}$ in the marine boundary layer, about one-fifth of the total $CCN_{0.2\%}$. This may be because the boundary aerosol nucleation scheme is based on organic-mediated nucleation (Metzger et al., 2010), limited by the concentrations of sulphuric

acid and SOA in the marine boundary layer. The difference between the total and boundary layer aerosol nucleation can be used to indicate the contribution to $CCN_{0.2\%}$ from free and upper troposphere aerosol nucleation. Therefore, the contribution of free and upper troposphere aerosol nucleation to the $CCN_{0.2\%}$ in the SEA marine boundary layer is about 41 %, which is consistent with the findings of Merikanto et al. (2009), who found that 45 % of the global marine boundary layer $CCN_{0.2\%}$ was contributed by nucleation in the free troposphere.

In terms of emission sources, anthropogenic emissions are the largest source of $CCN_{0.2\%}$ within the marine boundary layer, accounting for ∼ 26 % and ∼ 21 % of $CCN_{0.2\%}$ in the annual and BB seasonal averages, respectively. BB is the second largest contributor to $CCN_{0.2\%}$ within the marine boundary layer from emissions, accounting for ∼ 17 % and ∼ 19 % of $CCN_{0.2\%}$ in annual and BB seasonal averages. Although BB aerosols strongly influence this region, the contribution of anthropogenic

sources to the $CCN_{0.2\%}$ remains higher within the boundary layer. This may be due to $SO_2$, emitted from anthropogenic sources, which can increase $CCN_{0.2\%}$ by aerosol nucleation. Since aerosol nucleation is an essential source of $CCN_{0.2\%}$, nucleation of aerosols due to $SO_2$ from anthropogenic sources may be one of the main ways in which anthropogenic sources increase $CCN_{0.2\%}$. The $CCN_{0.2\%}$ contributed by sea salt, and DMS is mainly concentrated within the marine boundary layer, with ∼ 8 % and ∼ 16 % respectively in the annual mean.

The importance of BB and anthropogenic emissions to $CCN_{0.2\%}$ increases significantly in the cloud and plume layers. Both BB and anthropogenic emissions are transported from the African continent. Due to the convection over the land and the difference in altitude between the land and the ocean, these emissions are transported in the free troposphere above the cloud layer; therefore, BB and anthropogenic aerosol concentrations increase with altitude and subsequently their contribution to $CCN_{0.2\%}$.

In the cloud layer, BB contributes more to $CCN_{0.2\%}$ than anthropogenic emissions and is the largest source of $CCN_{0.2\%}$ in terms of emission sources. During BB season, BB contributes 43% of the $CCN_{0.2\%}$ in the cloud layer, even more than that from total aerosol nucleation, making BB the most significant source of $CCN_{0.2\%}$ overall (Fig. S3). The contribution of BB to $CCN_{0.2\%}$ further increases in the plume layer, with BB becoming the most dominant source of $CCN_{0.2\%}$ overall. The contribution of BB

to CCN$_{0.2\%}$ in the plume layer is 64 % on annual average, and increases to 76 % during BB season. This result highlights the significant impact of BB aerosols on CCN$_{0.2\%}$, especially during the BB season. However, as most of the CCN$_{0.2\%}$ contributed by the BB is distributed in the free troposphere, its effect on clouds is likely to be limited and similar to that of anthropogenic sources.

## 3.2 Cloud adjustments

### 3.2.1 Maximum supersaturation

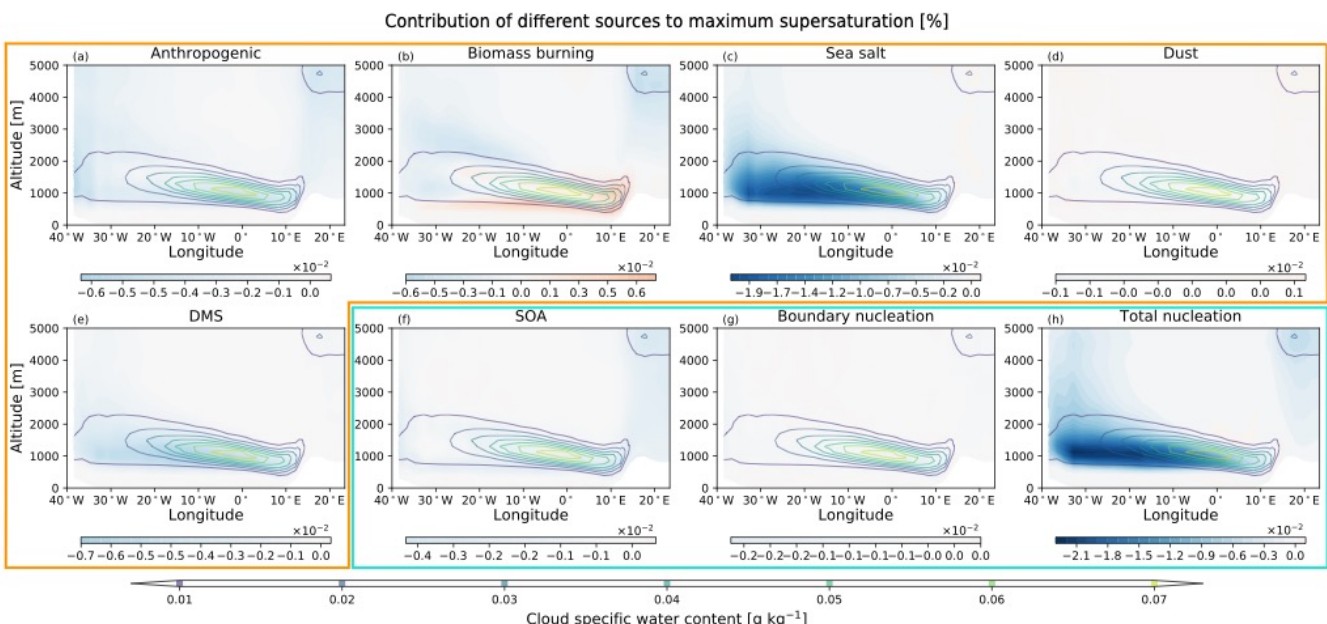

Figure 6. UKESM1 simulated annual mean vertical profiles of maximum supersaturation (%) from different sources. Profiles are averaged along the latitudes of the cloud box. The contributions of different sources to maximum supersaturation are listed in (a) to (h), where the contribution of emissions is shown in the yellow frame, and the contribution of atmospheric processes is shown in the light blue frame. The contour lines in each subplot are the cloud specific water content from the baseline simulation at the same temporal and spatial average. The same colourmap scale is used in each subplot to facilitate comparison, but the range differs in each plot, corresponding to the maximum and minimum of the maximum supersaturation.

As shown in Fig. 6, among the aerosols from various sources (emissions and atmospheric processes), BB aerosol is the only one that noticeably increases the maximum supersaturation. The increase in maximum supersaturation due to BB aerosol is more evident during the BB season, at approximately 0.028 % (Fig. S4). By contrast, all other aerosols generally exhibit a decreasing effect on the maximum supersaturation. The increase in maximum supersaturation due to BB aerosols is caused by the dynamical feedback due to shortwave absorption. Since most BB aerosols are located directly above the inversion layer,

their shortwave absorption can warm the surrounding air and enhance the underlying inversion. As a result, dry air entrainment is reduced and water vapour within the boundary layer is preserved, leading to an increase in maximum supersaturation, consistent with the findings in Che et al. (2021). Whereas for other types of aerosols, their effect on the maximum supersaturation is mainly through microphysical processes, i.e., acting as CCN. These aerosols, therefore, provide condensation sinks for water vapour, resulting in a reduction of the maximum supersaturation. Thus, as the largest contributor to $CCN_{0.2\%}$ in the marine boundary layer, total aerosol nucleation has the strongest effect on reducing maximum supersaturation among others. The decrease in maximum supersaturation due to sea salt is also apparent, second only to the effect of total aerosol nucleation. However, the annual mean $CCN_{0.2\%}$ number concentration contributed by sea salt is low in the marine boundary layer, only accounting for one-sixth of that from total aerosol nucleation. This is because, despite the low number concentrations, sea salt particle has a large diameter and therefore provides a larger surface to allow more water vapour to condense.

### 3.2.2 Cloud droplet number concentration

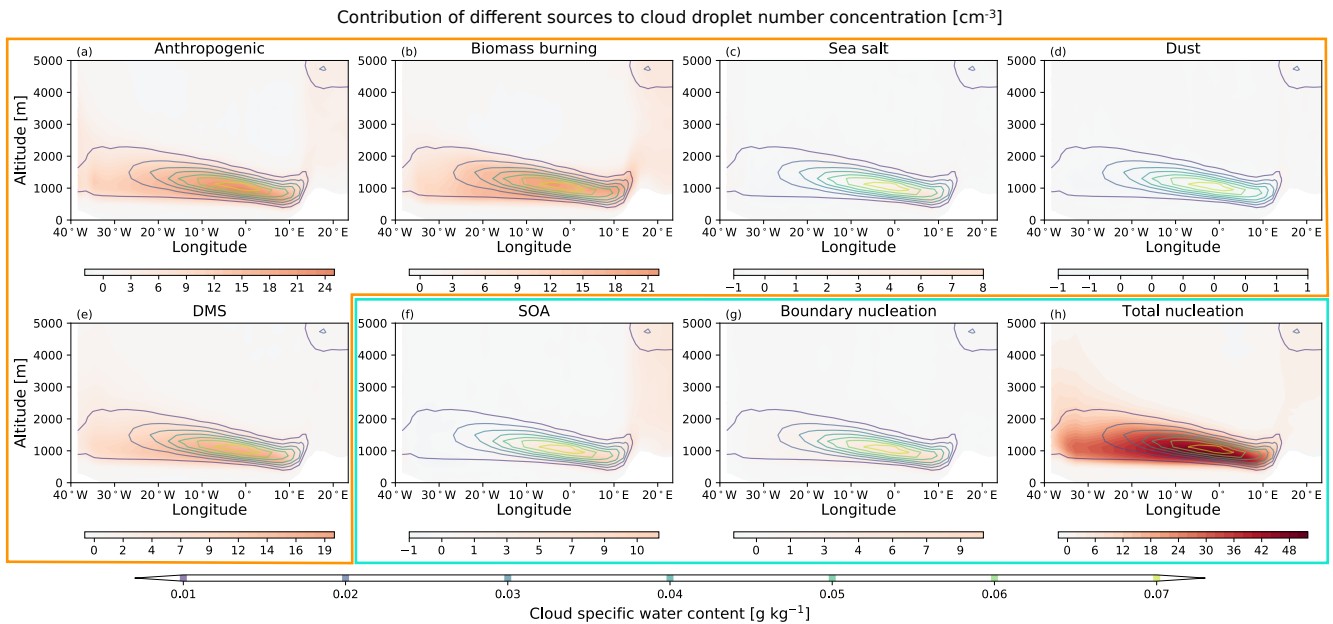

Figure 7. UKESM1 simulated annual mean vertical profiles of cloud droplet number concentration (CDCN) from different sources. Profiles are averaged along the latitudes of the cloud box. The contributions of different sources to CDCN are listed in (a) to (h), where the contribution of emissions is shown in the yellow frame, and the contribution of atmospheric processes is shown in the light blue frame. The contour lines in each subplot are the cloud specific water content from the baseline simulation at the same temporal and spatial average. The same colourmap scale is used in each subplot to facilitate comparison, but the range differs in each plot, corresponding to the maximum and minimum of the CDNC.

The annual means of cloud droplet number concentration (CDNC) from different sources are illustrated in Fig. 7. As can be seen from the figure, in general, the dominant source of CDNC is total aerosol nucleation, consistent with the source attribution of $CCN_{0.2\%}$ within the marine boundary layer. Previous studies have found that more than half of the CCN in the global marine boundary layer is contributed by aerosol nucleation (Clarke et al., 2013; Merikanto et al., 2009a; Williamson et al., 2019; Clarke and Kapustin, 2002), consistent with our result. However, source attribution in multiple models is recommended to confirm the importance of aerosol nucleation to the CDNC, as the nucleation and Aitken mode aerosol concentrations are significantly overpredicted by HadGEM models (Ranjithkumar et al., 2021; Gordon et al., 2020; Hardacre et al., 2021; Bellouin et al., 2013), suggesting the contribution from nucleation to CDNC may also be overestimated in our model. Even during the BB season, the concentration of CDNC contributed by total aerosol nucleation is similar to that of BB (Fig. S5), indicating that total aerosol nucleation remains the most significant source of CDNC throughout the years. In terms of emission sources, anthropogenic emissions make the highest contribution to the annual mean CDNC, slightly higher than the contribution of BB. This finding is also consistent with the result that anthropogenic contribute the highest proportion of $CCN_{0.2\%}$ of all emission sources in the marine boundary layer. BB contributes the second-largest annual mean of CDNC in terms of emission sources, closely followed by the contribution from DMS, which is consistent with their contribution to $CCN_{0.2\%}$ within the marine boundary layer. However, during the BB season, the importance of BB to CDNC increases significantly, contributing about the same amount of CDNC as total aerosol nucleation and almost twice as much as the anthropogenic emission (Fig. S5). The contribution of BB to CDNC during the BB season is higher than its contribution to $CCN_{0.2\%}$ within the boundary layer. This inconsistency is mainly due to the different contribution mechanisms of BB aerosols to CDNC compared to other aerosols. BB aerosols not only can provide CCN to increase CDNC, but also increase CDNC by influencing the vertical distribution of temperature through shortwave absorption, which in turn increases the maximum supersaturation in clouds (Che et al., 2021). This is also evidenced by Fig. 6. As a result, BB becomes the most important emission source of CDNC during the BB season. This result is also supported by a satellite study that found a clearly elevated CDNC with the presence of BB aerosols in this region (Redemann et al., 2021).

### 3.2.3 Liquid water path

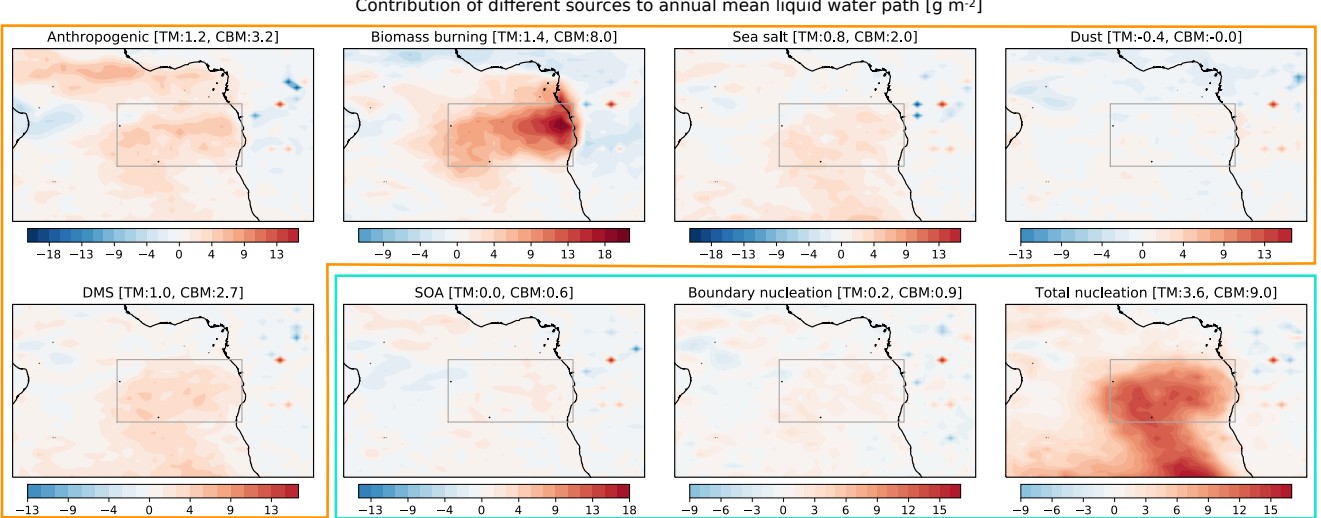

Contribution of different sources to annual mean liquid water path [g m-2]

Figure 8. UKESM1 simulated the annual mean liquid water path (LWP) from different sources. The contributions of different sources to LWP are listed in (a) to (h), where the contribution of emissions is shown in the yellow frame, and the contribution of atmospheric processes is shown in the light blue frame. The domain in each subplot ranges from 30° S to 10° N, and from 40° W to 30° E. The TM is the total mean of the domain, and the CBM is the mean of the cloud box (the grey box on the map). The same colourmap scale is used in each subplot to facilitate comparison, but the range differs for each plot, corresponding to the maximum and minimum of the LWP.

This section examines cloud adjustments due to different sources of aerosols, with a focus on LWP. From Fig. 8, LWP corresponds well with the CDNC in general for different sources. For sources having an apparent increase in CDNC, they also exhibit an increase in LWP. However, the ratio of the increased LWP to increased CDNC is different for those sources due to different aerosol properties. For the BB source, although the increased CDNC has a similar magnitude to that from anthropogenic emissions and is around half of that from total aerosol nucleation, the amount of LWP increased by BB in the cloud box region is slightly lower than that increased by total aerosol nucleation, and is nearly three times of that from the anthropogenic sources. This can be attributed to the radiative effect of BB aerosol, strengthening existing temperature capping inversion and reducing entrainment of sub-saturated air from above (Che et al., 2021; Deaconu et al., 2019; Sakaeda et al., 2011; Wilcox, 2010, 2012), thus increasing LWP. Sea salt shows a comparable (slightly lower) increase in LWP to that of anthropogenic and DMS sources in the cloud region, although its contribution to CDNC is much lower than that of anthropogenic and DMS emissions. This is probably due to the high hygroscopicity of sea salt aerosols, which allows them to uptake a large amount of water vapour above certain relative humidity and retain it in the form of liquid in the particles. Other

sources such as dust, SOA, and boundary layer aerosol nucleation only contribute a small amount of CDNC; therefore, the corresponding LWP increased by those sources may also be limited.

During the BB season, BB significantly increases the LWP within the cloud box region (21.7 g/m$^2$) and has the greatest impact on the LWP of all sources (Fig. S6). The amount of enhanced LWP by BB is two times that by total aerosol nucleation in the cloud box region, even with the similar amount of CDNC contributed by those two sources during the BB season. The higher LWP caused by BB when they are located directly above the inversion layer reflects the critical role of the radiative effect of BB aerosol in affecting cloud properties, and is consistent with our previous finding (Che et al., 2021).

### 3.3 Radiative effects

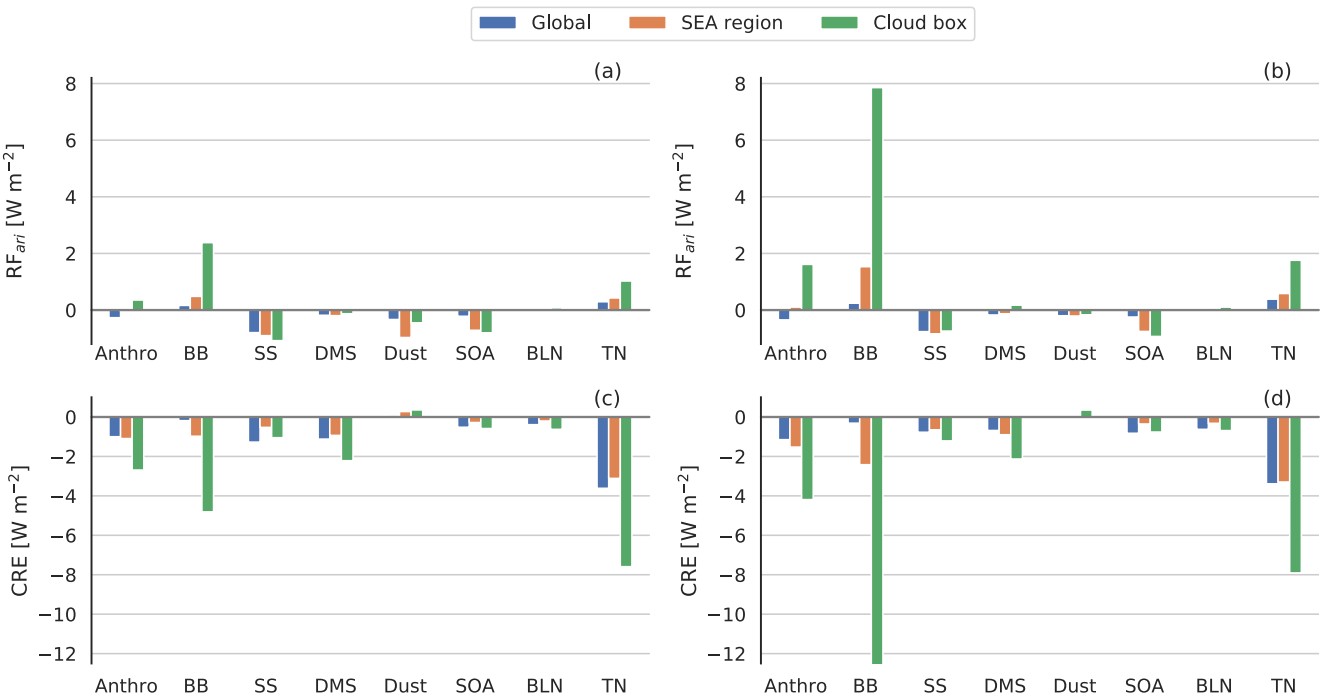

Figure 9. UKESM1 simulated radiative forcing from aerosol-radiation interaction (RF$_{ari}$) and cloud radiative effect (CRE) for different sources. (a) and (b) are the annual and BB season means of RF$_{ari}$, (c) and (d) are the annual and BB season means of CRE. Antro, BB, SS, DMS, Dust, SOA, BLN, and TN represent sources of anthropogenic, BB, sea-salt, DMS, dust, SOA, boundary layer nucleation, and total aerosol nucleation, respectively. Blue, orange, and green colours represent the mean values averaged in the global, investigated SEA area, and cloud box region, respectively.

The radiative effects of different sources are investigated in this section. The radiative forcing from aerosol-radiation interactions (RF$_{ari}$) and the cloud radiative effect (CRE) from aerosol-cloud interactions are calculated using the method of Ghan (2013). CRE includes rapid adjustments from aerosol-radiation interactions (known as aerosol semi-direct effect), and the effective radiative forcing from aerosol-cloud interactions (ERF$_{aci}$). As the aerosol semi-direct effect can impact the

temperature profile and further influence the cloud droplet size, number concentration, and cloud fraction, it is difficult to separate and reasonable to include it in the CRE. The effect of different aerosol sources on low-level cloud fraction is shown in Fig. S8 (annual mean) and S9 (BB seasonal mean). Overall, BB aerosols have the largest effect on low-level cloud fraction, increasing by 0.04 on an annual average and by 0.1 during the BB season in the cloud box region, followed by contributions from aerosol nucleation and anthropogenic sources. The increase in the cloud fraction from BB aerosols is mainly due to the

strengthening of the inversion layer by the shortwave absorption, which reduces dry air entrainment at the cloud tops and leads to an increase in the liquid water content of the clouds. In contrast, the increase in cloud fraction from anthropogenic emissions and the total aerosol nucleation process is driven by the increase in CCN and CDNC in the region due to aerosols from those two sources.

From Fig. 9, most aerosols exert a negative RF$_{ari}$ except for those from BB, anthropogenic, and total aerosol nucleation, especially in the cloud box region. As the sign of the RF$_{ari}$ depends on the relative brightness of the underlying surface and particles, RF$_{ari}$ of anthropogenic, BB, and total aerosol nucleation are positive in the cloud box region, as these aerosols are generally located above the cloud layer. This is more obvious for the BB season, during which the stratocumulus fraction and emissions from anthropogenic and BB increase coincidentally in the cloud box region. However, at the regional SEA and

global scales, only BB and total aerosol nucleation exhibit a warming RF$_{ari}$. This is because BB aerosols could absorb shortwave radiation and warm up the lower troposphere, while total aerosol nucleation produces a large number of small particles which can aggregate on the surface of the BC, thus increasing BC absorption through the coating. Sea salt exhibits the most notable cooling effect, and its RF$_{ari}$ shows little difference among global, the SEA, and the cloud box region. Although the aerosol concentration from sea salt is relatively low, the larger size of its particle makes sea salt the most crucial source of

aerosol radiative cooling.

Most aerosol sources show a negative effect from global to cloud box area for the cloud radiative effect. Total aerosol nucleation dominates the annual negative CRE, while during BB season, the most important source exerting negative CRE, especially in the cloud box region, is BB. This is consistent with the changes in LWP, as BB has brought a larger increase of

LWP during BB season than total aerosol nucleation. DMS shows negligible RF$_{ari}$ but comparable CRE to that of anthropogenic sources, consistent with the finding that it contributes a similar amount to LWP with the anthropogenic source. Combining RF$_{ari}$ and CRE, the effective radiative effect for each source is negative. The source showing the greatest total cooling is the total aerosol nucleation, though its RF$_{ari}$ is warming, confirming the critical role of aerosol nucleation in the low-level background cloud properties and the global radiation balance.

## Discussion and conclusion

In this paper, we use the United Kingdom Earth System Model (UKESM1) to attribute CCN and subsequent cloud property changes and radiative effects in the South-eastern Atlantic to different sources. The model has been evaluated with aircraft measurements from CLARIFY and ORACLES for the aerosol distribution, and is further evaluated in this study with LASIC in-situ observations for the marine boundary layer CCN. This framework guides our understanding of the effect of different aerosol sources (emissions and atmospheric processes) on marine stratocumulus clouds and radiation in the SEA.

From the results, overall, total aerosol nucleation is the most important source of $CCN_{0.2\%}$ in the marine boundary layer and cloud layer, both in terms of the annual and the BB seasonal means. In contrast, organic-mediated boundary layer nucleation contributes a much lower concentration of $CCN_{0.2\%}$, suggesting that it is not the main mechanism of CCN formation in the SEA region. This result highlights the importance of the free and upper troposphere nucleation and subsequent subsidence to aerosol number concentrations, which contribute $\sim 41$ % of $CCN_{0.2\%}$ in the SEA marine boundary layer. In terms of emissions, anthropogenic is the largest source of $CCN_{0.2\%}$ in the marine boundary layer, contributing an average of $\sim 26$ % and $\sim 21$ % of $CCN_{0.2\%}$ in the annual and BB seasons, respectively. The contribution of BB to $CCN_{0.2\%}$ in the marine boundary layer closely follows that of anthropogenic sources, at an average of $\sim 17$ % and $\sim 19$ % in the annual and BB season, respectively. Anthropogenic emissions contribute more $CCN_{0.2\%}$ than BB in the marine boundary layer, even during the BB season, which may be attributed to $SO_2$ emitted by the anthropogenic sources, as it can form aerosols through nucleation and thus provides more CCN. However, the importance of BB emissions to $CCN_{0.2\%}$ increases significantly in the cloud and plume layers. BB contributes $\sim 64$ % of the annual average $CCN_{0.2\%}$ in the plume layer, making it the most significant contributor for $CCN_{0.2\%}$. This result highlights the significant impact of BB aerosols on $CCN_{0.2\%}$, particularly in the region above the boundary layer. However, as most of the $CCN_{0.2\%}$ contributed by the BB is distributed in the free troposphere, its effect on clouds may still be limited by cloud-top entrainment. The contribution of aerosols from different sources to CDNC is consistent with their contribution to $CCN_{0.2\%}$ within the marine boundary layer in the cloud box region, highlighting the important role of boundary layer aerosols in clouds. Regardless of the annual or BB seasonal averages, total aerosol nucleation is the most dominant source of CDNC in general. In terms of emissions, anthropogenic sources are also the largest contributors to the annual average of CDNC, followed by BB. During the BB season, the contribution of BB to CDNC increases significantly (comparable to that of total nucleation to the CDNC), and is much higher than the contribution of anthropogenic sources to CDNC. This is mainly because BB aerosols, in addition to acting as CCN like anthropogenic aerosols, are generally located directly above the inversion layer and can enhance the underlying inversion layer through shortwave absorption, suppressing dry air entrainment at the cloud top and thus increasing the maximum supersaturation, leading to additionally increase in CDNC.

LWP generally corresponds well with the source attributed CDNC; however, the ratio of increased LWP to CDNC is different. With only half of the increased CDNC due to total aerosol nucleation, BB increases LWP by a similar amount as total aerosol

nucleation. The high ratio of LWP enhancement by BB emissions highlights the key role of the absorption of BB aerosol, enhancing the existing temperature inversion and reducing the entrainment of sub-saturated air. Sea salt also increases a more significant amount of LWP compared to CDNC, which may be due to the high hygroscopicity of sea salt particles. During the BB season, BB is the most important aerosol source increasing the LWP. Even though both sources contribute similar amounts of CDNC, the LWP increase in the BB is twice as large as the total nuclei in the cloud box region, indicating the key role of the BB aerosol radiation effect in affecting cloud properties. Anthropogenic emissions, BB, and total nucleation exert a positive warming $RF_{ari}$ in the cloud box region, as aerosols from those sources are mainly located above the clouds. Only aerosols from BB and total nucleation exert a positive $RF_{ari}$ in both the SEA and global, which is because BB aerosol could absorb shortwave radiation and warm the lower troposphere, while those small particles from total nucleation can aggregate on the surface of BC, thus increasing absorption through coatings. Sea salt shows the most notable negative $RF_{ari}$, although the aerosol concentration from sea salt is relatively low. For the cloud radiative effect, aerosol from all sources generally exhibits negative effects. Total aerosol nucleation dominates the annual-mean CRE perturbation, while during the BB season, BB dominates, consistent with the change of LWP. Combining $RF_{ari}$ and CRE, the effective radiative effect for each source is negative. The aerosol source showing the greatest total negative effect is the total aerosol nucleation, indicating the critical role of aerosol nucleation in modulating the background lower cloud properties and global radiation balance.

In our previous model evaluations, although the model is generally able to well simulate the horizontal and vertical distribution of aerosols in the SEA, aerosols are slightly underestimated at higher altitudes and are overestimated west of 5 °W(Che et al., 2021). The latter is also confirmed by studies with the same model (though with different configurations), which also showed an overestimation of aerosol concentrations in the western part of SEA (Gordon et al., 2020; Ranjithkumar et al., 2021). By comparing the modelled $CCN_{0.2\%}$ with observations, we find that although the model is generally in good agreement with the measurements, it still underestimates the peak $CCN_{0.2\%}$ during the BB season, suggesting that BB associated $CCN_{0.2\%}$ may be underestimated during the BB season. Also, when BC particles have a thick coating, the calculated κ may be underestimated by the volume mixing rule, which may further underestimate CCN concentrations associated with BB. These model biases introduce some uncertainties into our results, particularly with respect to the effects of BB aerosols on CCN and clouds. In addition, Doherty et al. (2022) showed that cloud cover is biased high in this region, at least for the 2017 biomass burning season, which could also lead to an overestimation of CRE. As a result, our results are subject to a certain level of model uncertainties. The discussion of different sources of CCN and their effects on clouds and radiation in this work is based on the averages during the BB season. However, from July to September, BB aerosol emissions vary with the burning conditions and areas, the marine boundary layer also evolves as the sea surface temperature decrease, and the stratocumulus cloud fraction also varies in different months. Therefore, the impacts of aerosol sources on CCN, clouds and radiation can be different for each month during the BB season, and require future studies. In addition, the influence of aerosols at different heights (boundary layer, cloud layer, free troposphere) on clouds and radiation is also an interesting issue that needs future investigation. The LASIC observational campaign can provide valuable continuous measurement data during the BB season

in 2016 and 2017, which can be used to validate the model's performance in the SEA marine boundary layer at a higher output resolution. ORACLES and CLARIFY aircraft observations can provide cloud and aerosol measurements at different altitudes, contributing to future studies of the effects of aerosols at different heights on clouds and radiation. The long-term LASIC observations also can provide sufficient data for the study of seasonal variation, benefiting future studies.

## 5   Data availability

The original model simulation data is available from JASMIN facility upon request.

**Author contributions**

PS and HC developed the concepts and ideas for the direction of the paper. HC and HG set up the model. HC carried out and analyzed the model simulation. DWP and HC performed the model validation, LD, DWP, HG, HC and PS contributed to the
analysis of the results. HC wrote the paper with input and comments from all other authors.

**Competing interests**

The authors declare that they have no conflict of interest.

**Special issue statement**

This article is part of the special issue "New observations and related modelling studies of the aerosol- cloud-climate system
in the Southeast Atlantic and southern Africa regions (ACP/AMT inter-journal SI)". It is not associated with a conference.

**Acknowledgements**

This research has been funded by the NERC CLARIFY project NE/L01355X/1. P.S. additionally acknowledges support from the NERC project NE/P013406/1 (A-CURE), the European Research Council (ERC) project constRaining the EffeCts of Aerosols on Precipitation (RECAP) and the H2020 project FORCeS under the European Union's Horizon 2020 research and
innovation program with grant agreements 724602 and 821205, respectively. HG acknowledges support from the NASA ROSES program under grant number 80NSSC21K1344. We sincerely acknowledge all CLARIFY and ORACLES science teams for data support. We acknowledge the use of the Monsoon2 system, a collaborative facility supplied under the Joint Weather and Climate Research Programme, a strategic partnership between the UK Met Office and the Natural Environment

Research Council (NERC). We also used the JASMIN facility (http://www.jasmin.ac.uk/) via the Centre for Environmental Data Analysis, funded by NERC and the UK Space Agency and delivered by the Science and Technology Facilities Council.

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
