# Peer review of "Source attribution of cloud condensation nuclei and their impact on stratocumulus clouds and radiation in the south-eastern Atlantic"

_Atmospheric Chemistry and Physics, 2022_

## Referee Comment (RC1)

**Review of "Source attribution of cloud condensation nuclei and their impact on stratocumulus clouds and radiation in the south-eastern Atlantic" by H. Che et al.**

This study examines the contribution of various emission sources and atmospheric processes to cloud condensation nuclei over the southeast Atlantic stratocumulus region and their impact on cloud properties, including droplet number concentration and liquid water path, and shortwave radiation using a large-scale modeling approach by turning off contributing components in the model one at a time. The manuscript is clearly written and easy to follow. I like the first part of the manuscript very much, i.e. source attribution of CCN, as it demonstrates nicely how does each source contributes to the overall CCN budget over the region, seasonally and annually, especially that these contributions are also examined as a function of the vertical structure of the emission sources and atmospheric processes. However, in the assessment of cloud adjustments and radiative impacts, this vertical structure aspect seems to be overlooked (or not included in the manuscript). Therefore, my main concern lies in the second half of the manuscript, and I suggest mostly minor revisions before being considered for publication.

Overall, I think this is a nicely designed and conducted study, which can make valuable contribution to the field. Here I provided some specific comments and suggested changes regarding my concerns of the manuscript.

**Major concerns/questions:**

1. The southeast Atlantic (at least the remote part) has a dynamic aerosol/smoke environment during the biomass burning season (*Zhang & Zuidema 2021 ACP*), such that early in the season (June-August), significant amount of biomass burning smoke is present in the marine boundary layer (*e.g. Zuidema et al. 2018 GRL, Zhang & Zuidema 2019 ACP*), whereas late in the season (September-October), as the southern African easterly jet builds up in the free-troposphere (*Adebiyi & Zuidema 2016 QJRMS*), smoke tends to be preferably located in the free-troposphere. This seasonal evolution in aerosol vertical structure can lead to different, even opposite, responses in cloud properties and SW radiation from month to month (*e.g. Zhang & Zuidema 2021 ACP*). In your 2021 ACP paper, you also pointed out the opposite signs of the semi-direct effect between remote and coastal SEA regions, in response to the different smoke vertical distributions.

   Although this study looked at annual and seasonal mean contributions/adjustments, I think it's worthwhile to look at contributions to cloud adjustments and radiative impacts from emission sources and atmospheric processes in different atmospheric layers, e.g. MBL, cloud layer, and FT, as you did for the CCN budget analyses.

   A question rather than a concern:
   Are you expecting (or not) to obtain heterogeneous attributions/adjustments/impacts on monthly scale (for BB at least) if you break your analyses into monthly means (as BB smoke vertical structure shifts from June to October), compared to the BB seasonal mean currently shown in the manuscript?

2. In the Method section, when I read the methodology part, i.e. how contributions from various emission sources and atmospheric processes are calculated, I assume they are calculated by simply taking the difference between baseline simulation and the runs with emissions/components turned off? I think this can be made clearer in the revised manuscript.

   Moreover, how well does this approach (emission/process turn-off) represent contributions from individual sources? For instance, as you pointed out, turning off anthropogenic emissions (or pre-industrial run) will also reduce nucleation processes (due to reduction in H2SO4 precursor), I wonder if there is way to quantify these entangled contributions and perhaps show that such entangled contributions are minimal/negligible compared to the individual ones.

3. Besides LWP and CDNC adjustments, do you also see cloud fraction changes attributed to these emission sources and nucleation processes. I think changes in cloud cover can also contribute to the CRE results you shown towards the end, as you also pointed out (P16, line 13). I tend to think it's worth showing cloud fraction changes in this study.

**Minor comments:**

P1, line 19, please make sure BB is defined when first used.

P2, line 24-26, I would argue that the radiative effect of BB aerosol can also act to reduce cloud fraction and LWP when smoke is present in the MBL (the cloud "burn-off" effect, e.g. *Zhang & Zuidema 2019 ACP, Ackerman et al. 2000 Science, Che et al. 2021 ACP*).

P11, line 23-24, please check this sentence, I see BB and anthropogenic contributions increase with altitude.

P13, line 19, 'through' → 'throughout'?

P14, line 6-7, is this not shown? I think this type of plot showing contributions from different atmospheric layers, i.e. vertical structure, is worth including.

P15, line 14-15, a rather minor point, just want to point out that an increase in CDNC does not always lead to an increase in LWP, as the sedimentation-entrainment and evaporation-entrainment feedbacks can decrease the LWP for non-precipitating stratocumulus (*e.g. Glassmeier et al. 2021 Science, Gryspeerdt et al. 2019 ACP*).

Figures 3, 5-7, these figures are nice; just a suggestion for the figure titles: perhaps adding the word 'contribution' to the end could help readers digest them faster. For me, I thought they represent absolute concentrations/SS/CDNC/LWP at first when I read the title, then I saw negative values on the color bars (seemed odd), then I realized that the values show differences between additional runs and the baseline run, which are representing contributions from individual sources.

---

## Author Response (AR1)

**Review replies to "Source attribution of cloud condensation nuclei and their impact on stratocumulus clouds and radiation in the south-eastern Atlantic" by Haochi Che et al.**

We would like to thank all reviewers for their constructive comments and suggestions on the manuscript. The feedback has allowed us to clarify and improve our manuscript.

The reviewer's comments are provided in **blue in the following**, and our responses are in **black**. Changes to the manuscripts made in response to the reviewer are in **green**. In addition, numerous changes in the manuscript are not shown in this response but are highlighted in the revised version.

**Reviewer #1:**

This study examines the contribution of various emission sources and atmospheric processes to cloud condensation nuclei over the southeast Atlantic stratocumulus region and their impact on cloud properties, including droplet number concentration and liquid water path, and shortwave radiation using a large-scale modeling approach by turning off contributing components in the model one at a time. The manuscript is clearly written and easy to follow. I like the first part of the manuscript very much, i.e. source attribution of CCN, as it demonstrates nicely how does each source contributes to the overall CCN budget over the region, seasonally and annually, especially that these contributions are also examined as a function of the vertical structure of the emission sources and atmospheric processes. However, in the assessment of cloud adjustments and radiative impacts, this vertical structure aspect seems to be overlooked (or not included in the manuscript). Therefore, my main concern lies in the second half of the manuscript, and I suggest mostly minor revisions before being considered for publication.

Overall, I think this is a nicely designed and conducted study, which can make valuable contribution to the field. Here I provided some specific comments and suggested changes regarding my concerns of the manuscript.

Major concerns/questions:

1. The southeast Atlantic (at least the remote part) has a dynamic aerosol/smoke environment during the biomass burning season (Zhang & Zuidema 2021 ACP), such that early in the season (June-August), significant amount of biomass burning smoke is present in the marine boundary layer (e.g. Zuidema et al. 2018 GRL, Zhang & Zuidema 2019 ACP), whereas late in the season (September-October), as the southern African easterly jet builds up in the free-troposphere (Adebiyi & Zuidema 2016 QJRMS), smoke tends to be preferably located in the free troposphere. This seasonal evolution in aerosol vertical structure can lead to different, even opposite, responses in cloud properties and SW radiation from month to month (e.g. Zhang & Zuidema 2021 ACP). In your 2021 ACP paper, you also pointed out the opposite signs of the semi-direct effect between remote and coastal SEA regions, in response to the different smoke vertical distributions. Although this study looked at annual and seasonal mean contributions/adjustments, I think it's worthwhile to look at contributions to cloud adjustments and radiative impacts from emission sources and atmospheric processes in different atmospheric layers, e.g. MBL, cloud layer, and FT, as you did for the CCN budget analyses.

We thank the reviewer's comment. The reviewer has raised two interesting questions. (1) Whether cloud adjustments and radiative forcings show variations in different months during the biomass burning season due to changes in the vertical structure of BB aerosols? (2) How do aerosols in the boundary layer, cloud layer, and free troposphere affect clouds and radiation?

**Q1: Whether cloud adjustments and radiative forcings show variations in different months during the biomass burning season due to changes in the vertical structure of BB aerosols?**

We investigated monthly variations in aerosol and cloud profiles during the BB season. Aerosols associated with BB and anthropogenic sources show a distinctive monthly variation from July to September. This is shown in Fig. R1 below. Aerosols from other sources do not have a distinct monthly variation.

[Figure]

Figure R1. UKESM1 simulated monthly mean profiles of aerosol concentration associated with BB and anthropogenic sources, with cloud specific water content. Profiles are averaged along the latitudes of the cloud box in the manuscript (Fig. 1). The same colourmap scale is used in each subplot to facilitate comparison, but the range differs for each plot, corresponding to the maximum and minimum of aerosol concentrations.

Aerosol concentrations from BB have the most significant monthly variation, with the highest in July and the lowest in September, showing the trend in BB emissions. The concentration of BB aerosols in the boundary layer (below the clouds) is also clearly higher in July than in the other two months. Aerosols from anthropogenic sources show a similar trend in the free troposphere, although the changes are not as significant as those of BB origin. However, there is no obvious

variation in the anthropogenic source aerosols in the boundary layer. Nevertheless, these monthly changes in the vertical distribution of aerosols from BB and anthropogenic sources during the BB season can result in cloud feedback and radiative forcing variations.

[Figure]

Figure R2. UKESM1 simulated monthly mean low-level cloud fraction from 2016 to 2017. The grey box (cloud box) in the map represents the region where low-level clouds dominate. The TM is the total mean of the domain, and the CBM is the mean of the cloud box. The same colourmap scale is used in each subplot to facilitate comparison, but the range differs for each plot, corresponding to the maximum and minimum of low-level cloud fraction in each month.

In addition to the vertical distribution of aerosols, there are also changes in the low-level cloud coverage. Fig. R2 shows the monthly variation of the low-level cloud fraction from the baseline simulation. In contrast to changes in BB aerosols, the proportion of clouds increases in the cloud box region from July to September. Since the direct radiative effect of aerosols depends on the relative albedo of the aerosol and underlying surface, changes in the distribution of both clouds and aerosols can therefore lead to changes in the direct radiative forcing of aerosols from July to September.

[Figure]

Figure R3. UKESM1 simulated monthly mean radiative forcing from (a) aerosol-radiation interaction ($RF_{ari}$) and (b) cloud radiative effect (CRE) for different sources in the cloud box region.

Anthro, BB, SS, DMS, Dust, SOA, BLN, and TN represent sources of anthropogenic, BB, sea-salt, DMS, dust, SOA, boundary layer nucleation, and total nucleation, respectively. Blue, orange, and green colours represent the mean values averaged in July, August, and September.

Fig. R3 shows the monthly variation of the aerosol-radiation interactions ($RF_{ari}$) and cloud radiative effect (CRE) from different sources. The most significant monthly variation in radiative forcing is from BB aerosol, which is also the aerosol with the highest radiative effect. $RF_{ari}$ associated with BB aerosol increases from July to August significantly (more than two times), corresponding to changes in the vertical distribution of BB aerosols and of cloud fraction. There is an apparent decrease in the concentration of BB aerosols in the marine boundary layer from July to August. Since BB aerosols in the marine boundary layer have a negative $RF_{ari}$, this change leads to an increase in $RF_{ari}$ of BB aerosols. In addition, despite the decrease in BB aerosols over the free troposphere, the apparent increase in the underlying cloud fraction may cause the subsurface of BB aerosols to become brighter, also leading to an increase in $RF_{ari}$. $RF_{ari}$ of BB aerosols decreases slightly from August to September, mainly due to the decline in BB aerosols within the free troposphere. However, $RF_{ari}$ associated with BB aerosols in September is still more than twice as high as in July, which suggests an important contribution from changes in the underlying clouds. Aerosol from anthropogenic and total nucleation have an increasing contribution to $RF_{ari}$ from July to September. As we discussed in the manuscript, aerosols from these sources are predominantly located above clouds; therefore, an increase in cloud fraction will lead to an increase in $RF_{ari}$. The negative $RF_{ari}$ for sea salt and SOA is due to these two types of aerosols being mainly within the marine boundary layer.

BB aerosols also show the strongest CRE and the most significant monthly variation. The CRE of BB aerosols decreases significantly from July to August, then increases from August to September, and the value is slightly larger in September than that in July. The inconsistency of the monthly variation in CRE of BB aerosols and low-level cloud fraction shows the role of the semi-direct effect of BB aerosols. There is a clear decrease in the free troposphere BB aerosols from August to September despite the increase in the cloud fraction. This can result in a decrease in the strong semi-direct cooling effect of BB aerosol, leading to an increase in CRE. (CRE include cloud adjustment and $RF_{aci}$).

Total nucleation has a decreasing trend of CRE (more negative) from July to August, which is related to the increase in the cloud fraction. We found in the manuscript that total nucleation is one of the main sources of CCN, thus, the increase in the cloud fraction also means that more cloud droplet is contributed from total nucleation, which is the reason behind the decrease in CRE.

**Q2: How do aerosols in the boundary layer, cloud layer, and free troposphere affect clouds and radiation?**

To quantitatively study the effect of aerosols from different layers (free troposphere, boundary layer, etc.) on clouds and radiation, models need to be able to trace track the aerosols at these different levels. Unfortunately, our model currently does not have this capability and therefore we cannot answer this question quantitatively. However, we can discuss this through the findings in the manuscript. For non-BB aerosols, their effect on clouds is mainly to act as CCN. Whereas we find in the manuscript that aerosols in the boundary layer are the ones that have the greatest effect

on cloud droplets, therefore we argue that for aerosols of non-BB origin, those within the marine boundary layer have the greatest influence on clouds. For BB aerosols, they have a poor hygroscopicity and are mainly located in the free troposphere, which hinders their ability to act as CCN and cloud droplets. However, due to their ability to absorb shortwave radiation, BB aerosols can enhance the inversion layer and reduce dry air entrainment, leading to an increase in the liquid water content of clouds. Thus, BB aerosols within the free troposphere have the greatest influence on clouds, which is supported by the strong increased LWP due to BB aerosols. For radiative forcing, the semi-direct effect of BB aerosols shows a predominant cooling, which is the result of the shortwave absorption by BB aerosols in the free troposphere, suggesting that free troposphere BB aerosols play a dominant role in affecting the radiation. For the other aerosols, the CREs are generally larger than the $RF_{ari}$, suggesting that the radiative effect of their interaction with clouds is more significant, indicating that for those types of aerosols, the concentration within the boundary layer is more important (act as CCN to affect cloud droplets).

Overall, the two questions are both interesting, but require further in-depth study. Since the manuscript focuses on the sources of CCN, we have revised it to briefly discuss these two issues and introduce the need for future work (P22, L19-27).

The discussion of different sources of CCN and their effects on clouds and radiation in this work is based on the averages during the BB season. However, from July to September, BB aerosol emissions vary with the burning conditions and areas, the marine boundary layer also evolves as the sea surface temperature decrease, and the stratocumulus cloud fraction also varies in different months. Therefore, the impacts of aerosol sources on CCN, clouds and radiation can be different for each month during the BB season, and require future studies. In addition, the influence of aerosols at different heights (boundary layer, cloud layer, free troposphere) on clouds and radiation is also an interesting issue and that need future investigation.

A question rather than a concern:

Are you expecting (or not) to obtain heterogeneous attributions/adjustments/impacts on monthly scale (for BB at least) if you break your analyses into monthly means (as BB smoke vertical structure shifts from June to October), compared to the BB seasonal mean currently shown in the manuscript?

Different months show changes in aerosol distribution/cloud distribution and radiative forcing. We have responded to this in reply to the previous comment.

2. In the Method section, when I read the methodology part, i.e. how contributions from various emission sources and atmospheric processes are calculated, I assume they are calculated by simply taking the difference between baseline simulation and the runs with emissions/components turned off? I think this can be made clearer in the revised manuscript.

Yes, the contributions are calculated by the difference between baseline simulation and runs with emissions turned off. We revised the manuscript as follows (P5L23-25, L31-32):

To facilitate our source attribution, four additional runs are made with BB, dust, sea-salt, and DMS emission turned off, and one simulation with pre-industrial CMIP6 emissions. The effect of these sources on aerosols, clouds and radiation can be derived from the differences between the baseline simulation and the individual runs with emissions turned off.

In addition to emission sources, three additional runs are performed without SOA (secondary organic aerosol) formation, boundary layer nucleation, and total nucleation, to allow attribution to atmospheric processes. The contributions of these atmospheric processes are also derived from differences between the baseline simulation and individual runs.

Moreover, how well does this approach (emission/process turn-off) represent contributions from individual sources? For instance, as you pointed out, turning off anthropogenic emissions (or pre-industrial run) will also reduce nucleation processes (due to reduction in H2SO4 precursor), I wonder if there is way to quantify these entangled contributions and perhaps show that such entangled contributions are minimal/negligible compared to the individual ones.

We thank the reviewer for the comment. Yes, changes in emissions of aerosols or their precursors can potentially induce non-linear responses of the global aerosol system, since such changes can affect the chemical and microphysical aging capacity of the atmosphere, resulting in a nonlinear response of aerosol populations with a coherence among the individual aerosol compounds. Stier et al. (2006) investigated the nonlinear responses of aerosols with emissions, and noticed such responses are mainly manifested in alterations of the aerosol lifetimes. However, the nonlinearities are generally not dominant and small compare to the effect of the emissions.

When comparing aerosols from emissions and atmospheric processes, there are some overlaps that cannot be neglected, since anthropogenic emission also includes nucleation precursors. This is why we discuss these two types of aerosol sources separately in the manuscript.

There is an overlap between anthropogenic emissions and nucleation (total nucleation) from atmospheric processes. To answer the reviewer's question, we have analysed aerosol concentration in the nucleation mode to find out how much overlap there is.

[Figure]

Figure R4 Average vertical profiles of nucleation-mode aerosol concentrations in the cloud box region simulated by UKESM1 during the 2016 and 2017 BB seasons. The blue and orange lines correspond to nucleation-mode aerosols from anthropogenic emissions (baseline – PI) and the nucleation process (baseline – nucleation-off ), respectively. The green line is the difference between them (nucleation – anthropogenic), roughly representing the nucleation without anthropogenic emissions.

From Figure R4, it can be seen that nucleation in this region occurs predominantly at an altitude of about 15000 m. After excluding anthropogenic emissions, nucleation-mode aerosol concentrations at this altitude decrease slightly (compare the orange and green lines), and this decrease accounts for ~ 10% maximum at this altitude. We therefore suggest that a maximum of around 10 % of the nucleation-mode aerosols are likely results from the anthropogenic precursors. Below 15000 m, anthropogenic sources also contribute larger aerosols, resulting in the coagulation of nucleation-mode particles, thus reducing the concentration of nucleation-mode particles at a lower altitude, which is why anthropogenic sources lead to a decrease in nucleation-mode particles at lower altitudes (negative values). Note that this is only a rough estimation.

The manuscript is revised according as follows: (P5L29-32, P6L3-4)

It should be noted that changes in emissions of aerosols or their precursors can affect the chemical and microphysical aging capacity of the atmosphere, resulting in a nonlinear response of aerosol populations. Stier et al. (2006) investigated the nonlinear responses and found they are generally not dominant and manifested in alterations of the aerosol lifetimes.

Precursors for nucleation include $H_2SO_4$, which is contained in both natural and anthropogenic emissions. Therefore, nucleation can also be affected by these emissions, particularly in areas with strong anthropogenic emissions (Saha et al., 2018). The exact extent of the impact of these emissions on nucleation remains unresolved in this work and requires future analysis.

3. Besides LWP and CDNC adjustments, do you also see cloud fraction changes attributed to these emission sources and nucleation processes. I think changes in cloud cover can also contribute to the CRE results you shown towards the end, as you also pointed out (P16, line 13). I tend to think it's worth showing cloud fraction changes in this study.

Thanks to the reviewer for the suggestion. We did see the variations of cloud fraction, although those variations are small. Fig. R5 shows the variation of annual mean cloud fraction due to different sources of aerosols.

[Figure]

Figure R5. UKESM1 simulated annual mean low-level cloud fraction from different sources in 2016 and 2017. The contributions of emissions are shown in the yellow frame, and the contribution of atmospheric processes is shown in the light blue frame. The domain in each subplot ranges from 30° S to 10° N, and from 40° W to 30° E. The TM is the total mean of the domain, and the CBM is the mean of the cloud box (the grey box on the map). The same colourmap scale is used in each subplot to facilitate comparison, but the range differs for each plot, corresponding to the maximum and minimum of the cloud fraction.

As seen in Fig. R5, BB aerosols have the greatest effect on cloud fraction, increasing it by an average of 0.04 within the cloud box region. This is followed by total nucleation and anthropogenic emissions. The increase in the cloud fraction from BB aerosols is mainly due to the strengthening of the inversion layer by the shortwave absorption, which reduces dry air entrainment at the cloud tops and leads to an increase in the liquid water content of the clouds. In contrast, the increase in the cloud fraction from anthropogenic emissions and the total nucleation process is driven by the increase in CCN and CDNC in the region due to aerosols from those two sources.

We have added the figure of cloud fraction in the supplement and briefly discussed its changes in the manuscript (P19L14-16, P20L1-5):

The effect of different aerosol sources on low-level cloud fraction are shown in Fig. S8 (annual mean) and S9 (BB seasonal mean). Overall, BB aerosols have the largest effect on low-level cloud fraction, increasing by 0.04 on an annual average and by 0.1 during the BB season in the cloud box region, followed by contributions from nucleation and anthropogenic sources. The increase in

the cloud fraction from BB aerosols is mainly due to the strengthening of the inversion layer by the shortwave absorption, which reduces dry air entrainment at the cloud tops and leads to an increase in the liquid water content of the clouds. In contrast, the increase in cloud fraction from anthropogenic emissions and the total nucleation process is driven by the increase in CCN and CDNC in the region due to aerosols from those two sources.

**Minor comments**

4. P1, line 19, please make sure BB is defined when first used.

Thanks for the reviewer, we have corrected this in the manuscript.

5. P2, line 24-26, I would argue that the radiative effect of BB aerosol can also act to reduce cloud fraction and LWP when smoke is present in the MBL (the cloud "burn-off" effect, e.g. Zhang & Zuidema 2019 ACP, Ackerman et al. 2000 Science, Che et al. 2021 ACP).

Yes, BB aerosols can reduce cloud fraction by heating the surrounding atmosphere and suppress the formation of the cloud droplets. However, this requires that BB aerosols are located mainly in the boundary layer, or within the clouds. BB aerosols in the southeast Atlantic are mainly located above clouds, with a low proportion within the boundary layer. We have previously analysed the radiative effect of BB aerosols (using the same model) and found that the radiative effect of BB aerosols in this region increases LWP, suggesting that the main effect of the radiative effect of BB aerosols in this region is to enhance clouds (show as in Fig. R6).

We therefore revised the manuscript as the below to avoid confusion (P2L25-27):

Previous studies suggest that the main role of the radiative effect of BB aerosol in the SEA is to strengthen the capping inversion and reduce the dry air entrainment from cloud tops, thereby increasing the LWP (as BB aerosols are mainly located above and near the inversion), and having a significant impact on the radiation balance (Ackerman et al., 2004; Deaconu et al., 2019; Gordon et al., 2018; Wilcox, 2010).

[Figure]

Figure R6. UKESM1 simulated BB aerosol radiative and microphysics effects on liquid water path during the BB season in 2016 and 2017. Figure from Che et al. (2021).

6. P11, line 23-24, please check this sentence, I see BB and anthropogenic contributions increase with altitude.

We have corrected this sentence, and changed the word "decrease" to "increase" (P12L22-25).

Due to the difference in altitude between the land and the ocean, the transport of these emissions is in the free troposphere above the cloud layer; therefore, BB and anthropogenic aerosol concentrations increase with altitude and subsequently their contribution to $CCN_{0.2\%}$

7. P13, line 19, 'through' → 'throughout'?

Thanks for the comment, we have corrected this in the manuscript.

8. P14, line 6-7, is this not shown? I think this type of plot showing contributions from different atmospheric layers, i.e. vertical structure, is worth including.

This is shown in Figure S5 in the supplement. We have basically only included annual averages in the manuscript as we intended to get an overall picture of CCN source attribution in the SEA. the BB season averages are all in the supplement.

9. P15, line 14-15, a rather minor point, just want to point out that an increase in CDNC does not always lead to an increase in LWP, as the sedimentation-entrainment and evaporation-entrainment

feedbacks can decrease the LWP for non-precipitating stratocumulus (e.g. Glassmeier et al. 2021 Science, Gryspeerdt et al. 2019 ACP).

Thanks for the comment, we have deleted this sentence. (P18L2-3)

10. Figures 3, 5-7, these figures are nice; just a suggestion for the figure titles: perhaps adding the word 'contribution' to the end could help readers digest them faster. For me, I thought they represent absolute concentrations/SS/CDNC/LWP at first when I read the title, then I saw negative values on the color bars (seemed odd), then I realized that the values show differences between additional runs and the baseline run, which are representing contributions from individual sources.

Thanks for the suggestion. We have updated all figures.

**Reviewer #2**

In this short study, the authors use the United Kingdom Earth System Model to attribute cloud condensation nuclei, supersaturation, cloud droplet number and liquid water path of stratocumulus clouds in the southeastern Atlantic to specific primary and secondary aerosols, and in particular biomass-burning aerosols (BBA). They find that BBA impact cloud droplet number and liquid water content through adjustments to aerosol-radiation interactions (absorption) rather than through aerosol-cloud interactions.

The paper is well written, with good figures. The main finding is not surprising, given that the BBA layer is rarely in contact with the cloud, but the story is worth telling.

I found two important aspects that would need clarifying:

1. The discussion does not distinguish between attribution of the baseline CDNC and LWP and the attribution of changes in those quantities. There is probably a Sc deck in the preindustrial simulation without BB. So BB/anthro cannot be the main drivers of LWP/CDNC, but they drive their temporal changes. Is that an accurate way of describing the findings?

Yes, although anthropogenic and BB emissions are two main sources of CCN in SEA (in terms of emissions), stratocumulus decks remain in the absence of anthropogenic and BB emissions as nucleation processes can contribute large amounts of CCN. In our response to reviewer#1 (Comment 2), we estimated that anthropogenic emissions contribute around 10% of nucleation-mode particles. Therefore the removal of anthropogenic emissions would have little impact on reducing CCN from nucleation. We stated that anthropogenic sources and BB are the main sources of CDNC from the point of view of emission sources. When emission sources and atmospheric processes are all considered, nucleation of aerosols is the most significant source of cloud droplets.

[Figure]

Figure R7. Annual average vertical profiles of CDNC in the cloud box region simulated by UKESM1 during 2016 and 2017. The blue, orange, green, and red lines are CDNC simulated from runs with baseline (PD), preindustrial (PI), BB emission off (noBBA), and nucleation off (noNuc).

As shown in Figure R7, the maximum annual mean CDNC concentration is ~ 97 cm$^{-3}$ at around 1100 m. When the anthropogenic emissions are switched off (PI), its concentration decreases to 78 cm$^{-3}$, a decrease of 20%. With BB emission switched off, the average annual CDNC is very similar to that with anthropogenic sources switched off, with a maximum annual average CDNC of 79 cm$^{-3}$, a decrease of about 19%. Although the maximum impact of these two sources on the annual CDNC is only 20%, it is the largest in terms of emission sources. However, for aerosols of atmospheric process origin, their impact on CDNC is more important. When the aerosol nucleation is switched off (with anthropogenic and BB sources on), the maximum annual mean CDNC concentration is 57 cm$^{-3}$, a decrease of about 41% from the baseline simulation, indicating the dominance of aerosol nucleation on cloud droplet concentrations.

Therefore, we believe that aerosol nucleation is the most important source of cloud droplets. However, particles contributed by nucleation are secondary aerosols generated by atmospheric processes. In terms of emission sources, we consider that anthropogenic and BB sources have the most important contribution among all other emission sources.

To avoid confusion, we have revised the manuscript accordingly to make this clear. (P16L5-14)

In general, the dominant source of CDNC is total nucleation, consistent with the source attribution of CCN$_{0.2\%}$ within the marine boundary layer. Previous studies have found that more than half of the CCN in the global marine boundary layer are contributed by nucleation (Clarke et al., 2013; Merikanto et al., 2009; Williamson et al., 2019; Clarke and Kapustin, 2002), consistent with our result. However, source attribution in multiple models is recommended to confirm the importance

of nucleation to the CDNC. Even during the BB season, the concentration of CDNC contributed by total nucleation is similar to that contributed by BB (Fig. S5), indicating that total nucleation remains the most significant source of CDNC throughout the years. In terms of emission sources, anthropogenic emissions make the highest contribution to the annual mean CDNC, slightly higher than the contribution of BB. This finding is also consistent with the result that anthropogenic contribute the highest proportion of $CCN_{0.2\%}$ of all emission sources in the marine boundary layer. BB contributes the second-largest annual mean of CDNC in term of emission sources, closely followed by the contribution from DMS

2. How do the findings depend on the way aerosol activation and cloud formation are represented in the model? It is important to insert a summary on Page 5 lines 4-5 of the representation of aerosol activation in the model and the calculation of cloud droplet number. Does that account for vertical transport in the boundary layer, or does it only consider those aerosols that happen to be in the same model level as liquid cloud water? And is cloud formation dependent on aerosols, or is liquid water content determined by thermodynamics, with droplet number being assigned in a second step? In other words, how close to the real world are the model and the attribution?

We thank the reviewer for these comments. We have revised the model description in the Method section to include details of CCN activation and cloud droplet formation (P5L2-10).

The bulk properties (cloud fraction, cloud liquid water content, etc.) of large-scale clouds are parameterized using the prognostic cloud fraction and prognostic condensate (PC2) scheme (Wilson et al., 2008a, b) with modifications described in Morcrette (2012). Cloud droplet activation is derived using the activation scheme of Abdul-Razzak and Ghan (2000). The activated CCN can be expressed as a function of aerosol properties (size, number, and composition) and thermodynamic properties (e.g., updraught velocity, temperature, pressure, and specific humidity), where thermodynamic properties are used to determine the local supersaturation, and aerosol properties are used to calculate activated CCN. When the local supersaturation has been determined, activated CCN is calculated with the $\kappa$-Kohler scheme, which uses a parameter $\kappa$ to represent the hygroscopicity of aerosols. The $\kappa$ value is set as 0.6, 0.2, and 1.2 for sulfate, organic, and sea-salt, respectively, and 0 for black carbon and dust (Engelhart et al., 2012; Petters and Kreidenweis, 2007). The internal volume mixing rule (Petters and Kreidenweis, 2007) is used to calculate the mean hygroscopicity of each mode. Cloud droplet concentrations at the cloud base are replicated vertically throughout contiguous columns of the cloud. After running the cloud activation scheme, the CDNC is then passed to the radiation and microphysics schemes.

3. The importance of nucleation in driving CDNC may also be a feature of Hadley Centre models. For example, Bellouin et al. 2013 https://doi.org/10.5194/acp-13-3027-2013 found a positive RFaci over pristine ocean regions, a feature that other AeroCom models (especially the ECHAM family, if I remember well) do not share. It could be worth giving a note of caution to that effect in the conclusion.

Thanks to the reviewer for pointing out this, we have added a note on the nucleation contribution to the CDNC, as suggested by the reviewer (P16L6-9).

The annual means of cloud droplet number concentration (CDNC) from different sources are illustrated in figure 6. As can be seen from the figure, in general, the dominant source of CDNC is total nucleation, consistent with the source attribution of $CCN_{0.2\%}$ within the marine boundary layer. Previous studies have found that more than half of the CCN in the global marine boundary layer are contributed by nucleation (Clarke et al., 2013; Merikanto et al., 2009; Williamson et al., 2019; Clarke and Kapustin, 2002), consistent with our result. However, source attribution in multiple models is recommended to confirm the importance of nucleation to the CDNC.

Other comments:

4. Page 5, line 10: Could clarify that the word anthropogenic is used here is the emission/CMIP sense. In a more general sense, most biomass burning emissions are anthropogenic too.

The manuscript was revised as suggested by the reviewer (P5L23-25).

The GFAS biomass burning and CMIP6 2014 emissions are used as the baseline simulation. To facilitate our source attribution, four additional runs are made with BB, dust, sea-salt, and DMS emission turned off, and one simulation with pre-industrial CMIP6 emissions. The effects of these sources on aerosols, clouds and radiation can then be derived from the difference between baseline simulation and individual runs with emissions turned off. The different aerosol sources (anthropogenic sources, biomass burning, etc.) are defined from the perspective of CMIP6.

5. Page 5, line 20. Are there units for the rate of 0.26?

No, this value is the fraction of how much monoterpene oxidation products can condense on particles to form SOA. 0.26 means 26%. The manuscript has revised as the follows (P6L9):

The gas-phase oxidations of Monoterpene by OH, $O_3$, and $NO_3$ yield SOA at a fixed rate of 0.26 (unitless, denotes 26% production percentage),

6. Page 7, figure 2: Is that total aerosol number or BBA only? I was expecting to see a secondary maximum near the surface, as stated on page 7 lines 14-16.

Figure 2 shows total aerosol number concentrations, as stated in the caption.

Due to the large particle size of sea-salt aerosols and their low number concentration, it is difficult to form a maximum near the ocean surface. Page 7 lines 14-16 mean boundary layer has low aerosol concentration, but it contains a larger fraction of sea-salt particles. It does not mean that the marine boundary layer has an aerosol peak due to sea salt.

We have modified the sentence to make it clearer (P8L6-7):

Therefore, the boundary is relatively clean, with the aerosol concentration around a few hundred per cubic centimetre. However, as the boundary layer is close to the sea surface, it contains a higher proportion of more hygroscopic sea-salt aerosols.

7. Page 11, 22-23: Isn't the free tropospheric transport more due to convection over land?

We have revised the manuscript as the following (P12L23-25):

Due to the convection over land and the difference in altitude between the land and the ocean, these emissions are transported in the free troposphere above the cloud layer.

8. Page 12, 14-15: BBA probably also has a decreasing effect on supersaturation via cloud droplet formation, which would offset some of the absorption-driven increase?

Yes, BBA can also decrease supersaturation by acting as CCN (microphysics effect). The effect of BBA on supersaturation has been investigated in detail in our previous work (Che et al., 2021), as is illustrated in Fig. R8. The microphysical effect of BBA can reduce supersaturation, but the reduction is smaller than the increase in supersaturation due to its radiative effect. Therefore, BB aerosols show an overall effect of increasing supersaturation, especially in regions close to the African continent.

In the manuscript, we find that only BB aerosols have the ability to increase supersaturation, and the paragraph in the manuscript is intended to explain the physical mechanism that causes the increase in supersaturation.

[Figure]

Figure R8. UKESM1 simulated BB aerosol radiative and microphysics effects on supersaturation during BB season in 2016 and 2017. Figure from Che et al. (2021).

9. Page 15, line 3: So stratocumulus evolution is driven by precipitation in the model? Are there non-precipitating Sc?

We have deleted this sentence.

10. Page 15, line 18: "times to that" --> "times that"

Thanks for this comment, we have corrected it.

**Reviewer #3:**

General overview

This paper describes a modelling study using UKESM to attribute the source of CCN in south east atlantic clouds and investigate the impact of the different CCN on the marine Sc and the associated radiation. Overall, the paper is OK, with clear figures and an obvious story throughout. It is general well written but some of the definitions and explanations need to be tightened up, so that it is clear to the reader where the definitions apply. As a result of this, I recommend this paper be accepted after minor revisions (see below)

Comments

1. Page 1, Line 15 - Define SEA (south east atlantic) in abstract

SEA was defined in the first sentence in the abstract.

2. Page 1, Line 17 - 18 - Could the authors quantify "the most to the annual average CCN…", i.e. what percentage?

Thanks for the comment, we have added the percentage. This sentence has been revised as the fellow (P1L19):

In terms of emission sources, anthropogenic emissions (from energy, industry, agriculture, etc.) contribute the most to the annual average $CCN_{0.2\%}$ in the marine boundary layer ($\sim 26$ %), followed by biomass burning (BB, $\sim 17$ %).

3. Page 1, Line 17 (and abstract) - Define BB (biomass burning)

We have corrected this in our response to the second reviewer.

4. Abstract - what is the definition of total nucleation and ensure the definition is consistent throughout the manuscript (See comment below for further information

Thanks for the suggestion. We have revised the manuscript to make it clearer (P1L17):

Overall, total nucleation (including tropospheric and stratospheric nucleation) is the dominant source of $CCN_{0.2\%}$ in the marine boundary layer.

5. Page 2, Line 25 to 27 - This statement is true when the BB airmass sits above and close to the inversion. It is not true when the BB mass is in the boundary layer or there is a gap between the absorbing air mass and the inversion. The authors need to add some text to clarify this statement, to avoid reader confusion.

Thanks for the suggestion. We have revised the manuscript to make it clearer as the follow (P2L25-28) .

Previous studies suggest that the main role of the radiative effect of BB aerosol in the SEA is to strengthen the capping inversion and reduce dry air entrainment from cloud tops, thereby increasing the LWP (as BB aerosols are mainly located above and near the inversion), and having a significant impact on the radiation balance (Ackerman et al., 2004; Deaconu et al., 2019; Gordon et al., 2018; Wilcox, 2010).

6. Page 4, line 25 - Add Walters et al reference for GA7.1, https://gmd.copernicus.org/articles/12/1909/2019/gmd-12-1909-2019.html

We have added the citation.

7. Page 4, line 29 and 30 - "The κ-Kohler activation scheme is implemented, which use a hygroscopicity parameter of each aerosol mode, κ, to calculate the activated CCN." From the present description, this differs from the standard Abdul-Razzak and Ghan scheme in the UKESM, which is fine but could the authors add some information about why they have made this change and what is the advantage/impact of this change. The authors refer to Che et al (2021) throughout the paper but Che et al uses a different activation scheme. Is this important?

We have modified this paragraph in our response to reviewer#2. κ-Kohler was used to set the hygroscopicity of the different aerosol species. In the standard UKESM/UM model, the hygroscopicity of aerosols is derived from the number of ions per dissociating molecule. Petters and Kreidenweis (2007) proposed a single parameter κ to characterise the hygroscopicity of aerosols, which has been widely used in observation studies since then. We used κ-Kohler and adopted the κ values from measurements in order to more realistically represent the CCN activation. The hygroscopicity of organic was 0 in default UKEMS1, here we have set the κ value for the organics to 0.2 to better match the observations (Che et al., in prep). Another reason why we use κ-Kohler is that the hygroscopicity of organics varies considerably with ageing, so we were planning to test the sensitivity of Sc clouds and radiation on different organic κ values in this area.

The activation scheme in this manuscript is essentially the same as that of Che et al. (2021), as ty they both based on Abdul-Razzak and Ghan (2000), the only difference being that the hygroscopicity in this manuscript is expressed in terms of κ, whereas in Che et al. (2021) it is derived from the ion number.

The paragraph has been revised as follows (P5L2-16).

The bulk properties (cloud fraction, cloud liquid water content, etc.) of large-scale cloud are parameterized using the prognostic cloud fraction and prognostic condensate (PC2) scheme (Wilson et al., 2008a, b) with modifications described in Morcrette (2012). Cloud droplet activation is derived using the activation scheme of Abdul-Razzak and Ghan (2000). The activated CCN can be expressed as a function of aerosol properties (size, number, and composition) and thermodynamic properties (e.g., updraught velocity, temperature, pressure, and specific humidity), where thermodynamic properties are used to determine the local supersaturation, and aerosol properties are used to calculate activated CCN. When the local supersaturation has been determined, activated CCN is calculated with the $\kappa$-Kohler scheme, which uses a parameter $\kappa$ to represent the hygroscopicity of aerosols. The $\kappa$ value is set as 0.6, 0.2, and 1.2 for sulfate, organic, and sea-salt, respectively, and 0 for black carbon and dust (Engelhart et al., 2012; Petters and Kreidenweis, 2007). The internal volume mixing rule (Petters and Kreidenweis, 2007) is used to calculate the mean hygroscopicity of each mode. Cloud droplet concentrations at cloud base are replicated vertically throughout contiguous columns of the cloud. After running the cloud activation scheme, the CDNC is then passed to the radiation and microphysics schemes for further calculations. The Coupled Model Intercomparison Project Phase 6 (CMIP6) emission data in 2014 is used (Eyring et al., 2016; Gidden et al., 2019). Due to the high interannual variability of BB emission, the global fire assimilation system (GFAS) version 1 data based on satellite fire monitoring is employed with a scaling factor of 2 (Johnson et al., 2016). A more detailed description can be found in our previous paper (Che et al., 2021).

8. Page 4, Line 29 - This work uses UKCA-mode dust, which differs from CLASSIC dust used in the UKESM. Similar comment to above, such a difference is fine but could the authors add some information about why they have made this change and whether this is important in the base simulations or when considering comparison with earlier work.

In the default UKESM/UM model, dust aerosols are indeed represented in the CLASSIC scheme, while other aerosols are from the GLOMAP-mode scheme. This setup is fine, but problems arise as we want to investigate the hygroscopicity of aerosols. Firstly it is difficult to adjust the hygroscopicity of dust aerosols, and secondly, it is difficult to calculate the degree of mixing of dust with other aerosols and the effect on the overall hygroscopicity of the mixed aerosols. The GLOMAP setting of dust also allows for condensation/coagulation on or with dust particles.

Pringle K. and Mann G. (https://www.ukca.ac.uk/images/3/3e/Dust_summary_180111.pdf) compared the CLASSIC dust and GLOMAP dust simulations in the UM model, and they concluded that GLOMAP is able to produce realistic accumulation mode dust loading, although the coarse mode loading is underestimated compared to the CLASSIC scheme. Since the coarse aerosols generally have a low concentration, we estimate that this bias will have a small impact on our results.

We have revised the manuscript to clarify this as follows (P5L2-4).

Differing from the standard configuration of representing the dust size distribution as six bins (Woodward, 2001), our configuration uses seven interactive log-normal aerosol modes in the microphysics scheme GLOMAP (Mann et al., 2010), comprising sulfate, sea salt, black carbon,

organic carbon and dust, allowing for condensation and coagulation on or with dust. With this setting we can set the hygroscopicity of different aerosol species with a single parameter $\kappa$.

9. Page 5 - Source attribution is achieved by switching off the emissions of BB, sea-salt, dust and DMS, respectively. While I understand why this has been done, how does switching off these emissions impact the simulation of cloud in these sensitivity tests. At present, this paper only shows cloud from the base simulations, so it is not possible to see what the impact of the changes in emissions are. Also, in switching of the emissions, I assume that the aerosol size distribution is impacted due to the removal of mass (and number). If so, does the change in size distribution impact the results or the conclusions? Does the change in size distribution influence any competition for vapour or the altitude at which water vapour is condensed?

Thanks to the reviewer for the comment. There are two main questions from this comment:

**Q1.** How does switching off these emissions impact the simulation of the cloud. At present, this paper only shows cloud from the base simulations, so it is not possible to see what the impact of the changes in emissions is.

The cloud fraction from the baseline simulation is shown on page 5. The effect of different aerosol sources on clouds is represented by the difference between the baseline simulation and the run with emissions switched off. Figs 6, 7 and 14 in the manuscript show the impact of those aerosol sources on cloud maximum supersaturation, CDNC, and LWP. Here we have added baseline results for supersaturation, CDNC, and LWP to the supplement to provide an overview of the baseline clouds.

[Figure]

Figure S2. UKESM1 simulated annual mean vertical profiles of maximum supersaturation, cloud droplet number concentration (CDNC), and annual mean of the cloud liquid water path (LWP). The TM is the total mean of the domain, and the CBM is the mean of the cloud box (the grey box on the map).

**Q2.** Also, in switching of the emissions, I assume that the aerosol size distribution is impacted due to the removal of mass (and number). If so, does the change in size distribution impact the results or the conclusions? Does the change in size distribution influence any competition for vapour or the altitude at which water vapour is condensed?

Yes, the aerosol size distribution is influenced by the emissions. Our study of the CCN focused on the CCN at a supersaturation of 0.2%, which is the total CCN obtained by integrating from above the critical activation diameter. Any variation in the aerosol size distribution is already included in this value. This means that we did not discuss in detail the changes in the aerosol number size distribution caused by changing the emission source (our main goal is to discuss the source attribution of CCN), but our conclusions include the effect of this aspect on CCN and clouds.

However, we did analyse the trends in CCN at other supersaturations in a previous draft version. The CCN at different supersaturation levels is strongly influenced by the aerosol number size distribution (and also by the hygroscopic capacity), so this analysis provides an indication of the role of the aerosol distribution on the CCN.

[Figure]

Figure R9. UKESM1 simulated (a-c) annual mean and (d-f) BB season mean concentrations of cloud condensation nuclei from different sources in the cloud box region as a function of supersaturation. (a) and (d) are averaged in the marine boundary layer in the cloud box region; (b) and (e) are in the cloud layer; (c) and (f) are in the plume layer. ANT, BB, SS, DMS, Dust, SOA, BLN, and TN represent sources of anthropogenic, BB, sea-salt, DMS, dust, SOA, boundary layer nucleation, and total nucleation, respectively. Note the scale of CCN concentration of (f) is different to facilitate a better visualization.

For BB and sea-salt emissions, their effect on CCN shows increasing with the supersaturation first, while after reaching the peak, the concentration of CCN starts to decrease with the increasing supersaturation. The decrease happens to relatively small diameters, such as nucleation and Aitken modes, and corresponds to the decrease of aerosol concentration in these modes. This means the BB and sea salt generally reduced aerosols in nucleation and Aitken modes. This may be because particles from these sources can serve as the condensation sink, the nucleation ratio is therefore reduced by decreasing in total sulphuric acid vapour.

For the absorption of water vapour, the chemical composition of the aerosol and the size of the particles both have an influence. When RH < 100%, the absorption of water vapour by the aerosol depends mainly on the chemical composition of the aerosol and not on the particle size (Zieger et al., 2010). When RH >100 %, i.e. in a supersaturated environment, the larger the particle size, the greater the ability of the aerosol to activate and allow water vapour to condense. This is reflected in the increased concentration of CCN at smaller supersaturation levels. From Fig. R9, aerosols from BB, sea-salt, anthropogenic, and DMS sources all increase the CCN at the low supersaturation (supersaturation less than 0.2%), indicating that water vapour is more likely to condense on these aerosols. The effect of BB and anthropogenic sources on CCN becomes more pronounced from the boundary layer to the plume layer, while other sources do not change much. It is important to note, however, that this does not mean that water vapour is more likely to condense on aerosols from BB and anthropogenic sources within the free troposphere. Condensation of water vapour requires not only particulate matter, but also supersaturation. The low relative humidity in the free troposphere does not allow water vapour to condense on the surface of aerosols and grow into cloud droplets. The effect of aerosols on water vapour in the free troposphere is mainly hygroscopic growth, i.e. it is mainly dependent on chemical composition rather than particle size.

Overall, the effect of aerosols on water vapour is complicated, and beyond the scope of our discussion of CCN sources, so it will not be discussed in this manuscript.

10. Page 5 - the definition of "total nucleation" is confusing. On page 5, it is defined as the sum of boundary layer and binary nucleation, while in the abstract it is defined as "total nucleation (binary nucleation)". Then on page 9 (line 15) the authors state that "Total nucleation contributes more to CCN0.2% compared to boundary layer nucleation, indicating a contribution from the free troposphere". So what is total nucleation? Could the authors clarify what they mean and ensure consistency.

We thank the reviewer for this comment. As we introduced in the method section, the boundary nucleation scheme is based on the organic-mediated nucleation (Metzger et al., 2010), determined by the concentrations of sulfuric acid and SOA, and limited to the boundary layer. The total nucleation includes the boundary layer nucleation and homogeneous binary nucleation of sulphuric acid and water, which is applicable to both tropospheric and stratospheric conditions, as described in Vehkamäki et al. (2002). Total nucleation thus includes boundary nucleation and free troposphere, and stratospheric nucleation.

We have revised the manuscript to provide a consistent description of total nucleation

11. Page 7 line 14 - add "layer" after boundary

We have corrected it.

12. Page 8 - Figure 3, why is the scale for ccn concentration from dust negative? Is this correct? If so could the authors explain what is going on?

Yes, dust indeed caused some decrease in CCN concentration at 0.2% supersaturation. We have plotted the effect of dust on CCN to make it more clear, as shown in Fig. R10.

[Figure]

Figure R10. UKESM1 simulated annual mean vertical profiles of CCN concentration at 0.2% supersaturation ($CCN_{0.2\%}$) from dust. Profiles are averaged along the latitudes of the cloud box. The contour lines are the cloud specific water content from the baseline simulation at the same temporal and spatial average.

The effect of dust on the CCN at 0.2% supersaturation is negligible, with a maximum increase of $\sim 8$ cm$^{-3}$ and a minimum decrease of $\sim 6$ cm$^{-3}$. The effect of dust on CCN is calculated by the difference between the baseline simulation and the dust removal run. A negative CCN means that there are more CCN when dust aerosols are removed.

In general, dust is considered to be mostly insoluble in water, so for activation it will increase the particle size (which may result in a lower critical supersaturation), but will not add ions to the solution (Raoul effect). When dust is mixed internally with other aerosol species, the average kappa value of the mixture is calculated by the volumetric mixing rule in the model, so the presence of dust will reduce the overall hygroscopicity, leading to a reduction in CCN. However, this is uncertain as other species may only coat the dust surface, which has little effect on hygroscopicity. This process is not represented in the global model and will require future investigation. Other effects may be that dust increases sulphuric acid condensation and reduces nucleation. As a result, CCN from nucleation is reduced by dust particles.

We have revised the manuscript as the follows (P10L25-26).

The reduction (negative) in CCN due to dust may be due to the increase in sulphuric acid condensation and the decrease in nucleation, thereby reducing the CCN from nucleation.

13. Page 9 line 3 to 7 - Do you see differing heating rates above the cloud between the simulations with and without BB? Also, does the cloud top height differ between these 2 simulations. It would

be useful to show such a difference since this will validate the authors speculation. At present Figure 3 only shows the baseline LWC, how do the clouds evolve in the sensitivity tests?

Fig. R11 shows the annual and BB seasonal mean temperature profile in the cloud box region. PD is a simulation of baseline and no-BB indicates that the BB emission source is switched off.

[Figure]

Figure R11. UKESM1 simulated annual and BB seasonal mean vertical profiles of temperature. Profiles are averaged in the cloud box region. Blue and orange lines are temperature (T) simulated from runs with baseline (PD), and BB emission turned off (no BB).

Fig. R11 clearly shows that BB aerosols enhance the inversion layer. When BB aerosols are present, the temperatures above 1500m are all higher than in the simulation without BB aerosols. This is more pronounced during the BB season, when BB aerosols increase substantially.

[Figure]

Figure R12. UKESM1 simulated annual and BB seasonal mean vertical profiles of cloud liquid water content. Profiles are averaged in the cloud box region. The upper panel (a-b) are BB seasonal averages, and the lower panel (c-d) are the annual means. PD represents the baseline simulation, and the no-BB represents the run with BB emission turned off.

The reviewers also asked whether cloud height varied with the presence of BB aerosols. As can be seen from Figure R12, the cloud tops are higher in runs where BB emissions are switched off, and this variation is particularly noticeable during the BB season. Thus, the presence of BB aerosols above the clouds does reduce cloud height, which is mainly caused by the changes in temperature profile due to the absorption of shortwave radiation by the BB aerosols (Fig. R11)

The final question, how clouds evolve, has been included in our supplement with the properties of baseline clouds and answered in our response to reviewer#3 question 9. (page 17 in the response).

14. Page 11 line 15 to 17 - "This may be due to SO2 emitted from anthropogenic sources, which can increase CCN0.2% by nucleation." The authors speculate here but they do not demonstrate this. Is there anything else that could cause this? If so, the authors should state this. Ideally, it would be good to show this, if it is possible, with another sensitivity run.

Firstly, anthropogenic emissions include SO2, which is a precursor for aerosol nucleation, so anthropogenic emissions are capable of increasing nucleation reactions. We believe that our explanation is reasonable and if the reviewer has other suggestions, we are happy to consider them. Secondly, the estimate of the contribution of anthropogenic emissions to nucleation is answered in the response to reviewer#1 comment 2. Finally, how anthropogenic emissions affect nucleation is beyond the scope of this report, which focuses on the source attribution of CCN.

15. Page 12 line 14 - 17 - "The increase in maximum supersaturation due to BB aerosols is caused by their shortwave radiation absorption effect. As it can warm the air due to its absorption of shortwave radiation, BB aerosol can enhance the inversion layer over clouds, preserving water vapour within the boundary layer and increasing the maximum supersaturation, consistent with the finding in Che et al. (2021)". This description is potentially misleading and confusing. In particular, "The increase in maximum supersaturation due to BB aerosols is caused by their shortwave radiation absorption effect", is not correct since the increase in boundary layer max supersaturation is caused by a dynamic feedback that results from the increased absorption. For example, Johnson et al (2004) demonstrated that an absorbing layer directly above the marine Sc deck will lead to an enhancement of the inversion strength, which will reduce the entrainment of warm and drier free troposphere air into the boundary layer. This leads to an increase in LWP compared to a simulation with no absorbing layer. In the work under review, the authors only focus on the preservation of the water vapour in the boundary layer and do not address the temperature profile. The impact of the BB layer on the inversion is referred to a lot but the authors do not present any profiles (potential temperature, vapour or RH) to demonstrate a strengthening of the temperature or moisture inversion. Could the authors add these to prove these statements about strengthening inversion?

We thank the reviewer for the comment. We have revised the manuscript to make it clear that the variation in supersaturation is resulted by the dynamical feedback that caused by the shortwave absorption of BB aerosols (P14L9-13).

The increase in maximum supersaturation due to BB aerosols is caused by the dynamical feedback due to short-wave absorption. Since most BB aerosols are located directly above the inversion layer, their short-wave absorption can warm the surrounding air and enhance the underlying inversion. As a result, dry air entrainment is reduced and water vapour within the boundary layer is preserved, leading to an increase in maximum supersaturation, consistent with the findings in Che et al. (2021).

For the temperature profile, we have shown the strengthening of the inversion by BB aerosol in Fig. R11 and Fig. R12. The strengthening of the inversion layer by BB aerosol in the SEA is discussed in detail in previously published papers (e.g., Gordon et al., 2018; Haywood et al., 2021; Zhang and Zuidema, 2019).

16. I appreciate that the authors refer to Che et al 2021 as the reference for the impact of absorbing aerosol over marine Sc and the supplemental plots in Che et al 2021 show this impact. However, the simulations presented in this work seem to use some different parametrisations, e.g. activation, dust, so the results may not be directly comparable. Also the description in Che et al is as follows, "Near the coast, BBAs are generally above the underlying cloud deck; the absorption aerosols could strengthen the boundary layer inversion (Fig. S4) and thus decrease the dry air entrainment, resulting in increased humidity and hence supersaturation". This is a better description than "preserves water vapour in the boundary layer", since it is the RH that matters.

We thank the reviewer for his meticulousness and for reading our previous work. Yes, the activation schemes in the two papers appear to be different, but they are essentially the same, both based on the scheme of Abdul-Razzak and Ghan (2000). The only difference is that the

hygroscopicity of the aerosol is expressed using kappa in this manuscript. We have revised the manuscript to make this clear and do not assume that it significantly affects our conclusions.

For dust aerosols, we responded and discussed the reviewer's previous question (comment 8). In summary, yes, the modelling of dust is different, but we would expect a negligible effect of dust because its concentration is very small in the SEA.

We have revised the manuscript to make this sentence clearer (P14L9-13).

The increase in maximum supersaturation due to BB aerosols is caused by the dynamical feedback due to short-wave absorption. Since most BB aerosols are located directly above the inversion layer, their short-wave absorption can warm the surrounding air and enhance the underlying inversion. As a result, dry air entrainment is reduced and water vapour within the boundary layer is preserved, leading to an increase in maximum supersaturation, consistent with the findings in Che et al. (2021).

17. Page 15, line 19 to 21 - "The higher LWP caused by BB reflects the critical role of the radiative effect of BB aerosol in affecting cloud properties, and is consistent with our previous finding (Che et al., 2021)." I think it is important to state that this critical role will only occur where the absorbing layer is directly above the inversion. If there is a gap between the absorbing aerosol layer and the cloud so that the absorbing layer does not impact the inversion then this response is not seen. This is demonstrated in Haywood, J. M., S. R. Osborne, P. N. Francis, A. Keil, P. Formenti, M. O. Andreae, and P. H. Kaye, The mean physical and optical properties of regional haze dominated by biomass burning aerosol measured from the C-130 aircraft during SAFARI 2000, J. Geophys. Res., 108(D13), 8473, doi:10.1029/2002JD002226, 2003

We thank the reviewer for the suggestion. We have revised the manuscript accordingly to emphasize that this role is mainly for BB aerosols located directly above the inversion layer (P18L20-21).

The higher LWP caused by BB when they are located directly above the inversion layer reflects the critical role of the radiative effect of BB aerosol in affecting cloud properties, and is consistent with our previous finding (Che et al., 2021).

18. Page 19, line 14 to 16 - "This is mainly because BB aerosol, in addition to acting as CCN like anthropogenic aerosol, also can increase the maximum supersaturation through the radiative effect of its shortwave absorption, thus additionally increasing the CDNC. "Again this comment is similar to the previous comment - I do not like this statement and I think it is misleading. BB aerosol does not increase the maximum supersaturation of the boundary layer. Instead, when BB aerosol is directly above the inversion the associated absorption will strengthen the inversion, reduce entrainment mixing of warm dry air from aloft, which will permit a higher RH and max supersaturation. If the BB aerosol is separated from the cloud layer or in the boundary layer then the associated absorption will lead to know change in the supersaturation or a decrease. Could the authors be more specific with this type of statement?

Thanks for the reviewer's comment. We revised this statement to make it more specific (P21L22-25).

[revised manuscript text omitted]

---

## Editor Decision (ED1)

**Editor's Report:**

Three reviewers have provided comments, with reviewer 1 suggesting minor revisions that account for the known variation in aerosol vertical structure from July-September, reviewer 2 also suggesting minor clarifications, primarily of the model and how it is used, and reviewer 3 also suggesting minor revisions, primarily related to the language.

After reviewing the comments, the authors' response, and the overall manuscript, it is my determination that the authors have not sufficiently responded to the intent of the reviewer comments. I suggest the authors give this another go-through. Reading through the manuscript, I also have several specific comments, listed below in primarily chronological order, that I would like to see addressed before the paper is finalized.

Abstract: Please read this over more carefully. Keep in mind many readers will not look past the abstract.

- 1. The description of 'total nucleation' in the abstract as 'tropospheric and stratospheric nucleation' is confusing. This is the first time the reader encounters this term. It's worth including an additional sentence here to define the term.
- 2. Mention the time period you are looking into.
- 3. Lines 21-23 seem to include two contradictory phrases perhaps the authors mean to say most of the BL CCN is introduced from above through entrainment? Wouldn't BB then be the dominant BL CCN source?
- 4. Line 25: the reader doesn't yet know the model simulation places most of the aerosol above the BL. Keep reviewer 3's comments in mind here and rewrite.
- 5. State something about the model aerosol vertical structure in the abstract results are highly dependent on the aerosol being located above the cloud.

6. The reader might be reasonably surprised to read that non-BB anthro emissions are the largest contributors to MBL CCN\_0.2%, above BB and sea spray. On p. 12 you clarify the CCN in the cloud layer - thus those most likely to form cloud droplets - are more likely to be BB. This would be worth mentioning in the abstract.

**Overall:**

The authors conclude, that because in the UKEMS1 model, the BBA is mainly in the FT for July-September, and has a poor hygroscopicity, that BBA is less important as a BL CCN, and instead, primarily serves to strengthen the inversion top. This is valuable to know about the UKESM1 model behavior. What is less clear is how well the UKESM1 simulations are capturing the observations. The reviewer comments relate to this: Reviewer 1 mentions the seasonal cycle. Rev 2 and 3 mention model characteristics.

For example: the LASIC campaign has shown that there can be significant CCN in the BL, for example, Zuidema et al 2018 Fig. 1, reproduced below, shows CCN\_0.2% reaching 103 /cc. The temporal variation indicates it is primarily modulated by BC. These values I believe exceed those shown in Fig. R1, though Fig. R1 is difficult to interpret; absolute values for the CCN depicted would have helped, or at least an explanatory caption. The July-September model means shown in Fig. 2 are difficult to interpret for the BL, and a 3-month model mean doesn't communicate the range.

In addition, in several portions of the manuscript, the authors refer to Che 2021, as a model validation paper. That paper only compared aerosol extinctions along CLARIFY and ORACLES flight tracks, with the ORACLES flight tracks spending little time in the boundary layer. During September, the LASIC values also indicate a clean MBL, consistent with ORACLES-2016, but a BL lacking smoke in September does not mean the BL is also non-smoky in July and August. The comparison to the aircraft flight track data isn't a sufficient validation for the 3 months, in contrast to the statement on p. 8, line14. Is the model genuinely capturing the boundary layer

smoke in July and August? Can the authors create a figure from their model simulation that is comparable to the LASIC data? The authors also refer to several other papers as a form of validation (e.g., p. 2, line28): Deaconu 2019 and Wilcox 2010 rely on satellite datasets that have difficulty distinguishing BB within the BL from sea-spray, Gordon 2018 is a modeling study focusing on a 10-day August time period only that produced an unrealistic 8K warming in the free troposphere, and Ackerman 2004 is for a different location. These references ignore the new information we have thanks to LASIC, ORACLES, and CLARIFY. Besides the studies mentioned by Rev 1, there are also more detailed StCU-to-Cu transition papers indicating the BB can also have a radiative impact in the BL.

Similarly, Kacarab et al. 2020 is also relevant, indicating a kappa of 0.4 for smoke based on oracles observations. This study should be referenced and discussed somewhere, as it does not support the low hygroscopicity for smoke assumed here. https://acp.copernicus.org/articles/20/3029/2020/.

---

## Author Response (AR2)

We would like to thank the editor for the thoughtful and detailed comments on our paper. We feel that in responding to these comments and suggestions, we have significantly improved the quality and readability of the paper.

The editor's comments are provided in **blue in the following**, and our responses are in **black**. Changes to the manuscripts made in response to the reviewer are in **green**. In addition, some changes in the manuscript are not shown in this response but are highlighted in the revised manuscript.

**Editor's Report:**

Three reviewers have provided comments, with reviewer 1 suggesting minor revisions that account for the known variation in aerosol vertical structure from July-September, reviewer 2 also suggesting minor clarifications, primarily of the model and how it is used, and reviewer 3 also suggesting minor revisions, primarily related to the language.

After reviewing the comments, the authors' response, and the overall manuscript, it is my determination that the authors have not sufficiently responded to the intent of the reviewer comments. I suggest the authors give this another go-through. Reading through the manuscript, I also have several specific comments, listed below in primarily chronological order, that I would like to see addressed before the paper is finalized.

Abstract: Please read this over more carefully. Keep in mind many readers will not look past the abstract.

1. The description of 'total nucleation' in the abstract as 'tropospheric and stratospheric nucleation' is confusing. This is the first time the reader encounters this term. It's worth including an additional sentence here to define the term.

We thank the editor for the suggestion. Total nucleation includes nucleation in and above the boundary layer. Our model results find that boundary nucleation has a small effect on marine boundary layer CCN (0.2%), probably because boundary nucleation is suppressed by the sufficient condensation sinks provided by the marine boundary layer aerosols (e.g., sea salt), or limited by the available organic aerosols. In contrast, particles nucleated above the boundary layer can grow and subside into the marine boundary layer, contributing to the majority of CCN (41 % on annual average), which is consistent with the findings of Merikanto et al. (2009), who found that 45 % of the global marine boundary layer CCN (0.2%) was contributed by the free troposphere nucleation.

In order to analyze the height at which nucleation predominantly occurs, we calculated 2016-2017 mean vertical distribution of nucleation modal aerosols in the cloud box region, as shown in Figure 1. As can be seen from the figure, the nucleation mode aerosols peak at 200hPa, indicating nucleation mostly occurs in the free and upper troposphere. Therefore, we consider the nucleation in the free and upper troposphere to be the dominant source of CCN in the boundary layer.

To make it clearer, we removed the term "total nucleation" from the abstract and emphasized nucleation in the free and upper troposphere. The sentence was modified as follows.

P1L16-18

Overall, free and upper troposphere nucleated aerosols are the dominant source of boundary layer $CCN_{0.2\%}$, contributing an annual average of $\sim 41$ % as they subside and entrain into the marine boundary layer, which is consistent with observations highlighting the important role of nucleation for boundary layer CCN.

[Figure]

Figure 1. Mean vertical distribution of nucleation mode aerosols in the cloud box region for 2016-2017, simulated by UKESM1. The black line is the mean distribution, and the blue-shaped area represents the standard deviation.

2. Mention the time period you are looking into.

We revised the manuscript as follows.

P1L14-16

In this paper, we use the United Kingdom Earth System Model to investigate the sources of CCN (from emission and atmospheric processes) in the SEA, and the response of cloud droplet number

concentration (CDNC), cloud liquid water path (LWP), and radiative forcing to those sources during 2016 and 2017.

3. Lines 21-23 seem to include two contradictory phrases - perhaps the authors mean to say most of the BL CCN is introduced from above through entrainment? Wouldn't BB then be the dominant BL CCN source?

We have removed this sentence. The revised text reads as follows.

Overall, free and upper troposphere nucleated aerosols are the dominant source of boundary layer $CCN_{0.2\%}$, contributing an annual average of ~ 41 % as they subside and entrain into the marine boundary layer. In terms of emission sources, anthropogenic emissions (from energy, industry, agriculture, etc.) contribute the most to the annual average $CCN_{0.2\%}$ in the marine boundary layer (~ 26 %), followed by biomass burning (BB, ~ 17 %). In the cloud layer, BB contributes about 34 % of annual $CCN_{0.2\%}$, midway between the contributions from aerosol nucleation (36%) and anthropogenic sources (31%). The contribution of aerosols from different sources to CDNC is consistent with their contribution to $CCN_{0.2\%}$ within the marine boundary layer, with free and upper troposphere aerosol nucleation being the most important source of CDNC overall. In terms of emission sources, anthropogenic sources are also the largest contributors to the annual average of CDNC, closely followed by BB.

4. Line 25: the reader doesn't yet know the model simulation places most of the aerosol above the BL. Keep reviewer 3's comments in mind here and rewrite.

We have revised the sentence as follows.

P1L25-27

The contribution of BB to CDNC is more significant than its increase to $CCN_{0.2\%}$, mainly because BB aerosols are mostly located directly above the inversion layer in the model, thus can increase CDNC by enhancing the maximum supersaturation through the dynamical feedback due to shortwave absorption.

5. State something about the model aerosol vertical structure in the abstract - results are highly dependent on the aerosol being located above the cloud.

We added the BB aerosol vertical information in response to comment 4.

We have made the following revision to L31-32 to emphasise the vertical distribution of BB aerosols.

P2 L2-4

However, as most BB aerosols are located directly above the inversion layer, their effect on clouds increase due to its absorption effect (about the same as anthropogenic sources for CDNC and more than anthropogenic sources for LWP), highlighting the crucial role of its radiative effect on clouds.

6. The reader might be reasonably surprised to read that non-BB anthro emissions are the largest contributors to MBL CCN_0.2%, above BB and sea spray. On p. 12 you clarify the CCN in the cloud layer - thus those most likely to form cloud droplets - are more likely to be BB. This would be worth mentioning in the abstract.

We added the following sentence in the abstract (L19-20)

In the cloud layer, BB contributes about 34 % of annual $CCN_{0.2\%}$, midway between the contributions from aerosol nucleation (36%) and anthropogenic sources (31%).

Overall:

The authors conclude, that because in the UKEMS1 model, the BBA is mainly in the FT for July-September, and has a poor hygroscopicity, that BBA is less important as a BL CCN, and instead, primarily serves to strengthen the inversion top. This is valuable to know about the UKESM1 model behavior. What is less clear is how well the UKESM1 simulations are capturing the observations. The reviewer comments relate to this: Reviewer 1 mentions the seasonal cycle. Rev 2 and 3 mention model characteristics.

[Figure]

For example: the LASIC campaign has shown that there can be significant CCN in the BL, for example, Zuidema et al 2018 Fig. 1, reproduced above, shows CCN_0.2% reaching 103 /cc. The temporal variation indicates it is primarily modulated by BC. These values I believe exceed those shown in Fig. R1, though Fig. R1 is difficult to interpret; absolute values for the CCN depicted would have helped, or at least an explanatory caption. The July-September model means shown in Fig. 2 are difficult to interpret for the BL, and a 3-month model mean doesn't communicate the range.

In addition, in several portions of the manuscript, the authors refer to Che 2021, as a model validation paper. That paper only compared aerosol extinctions along CLARIFY and ORACLES flight tracks, with the ORACLES flight tracks spending little time in the boundary layer. During September, the LASIC values also indicate a clean MBL, consistent with ORACLES-2016, but a BL lacking smoke in September does not mean the BL is also non-smoky in July and August. The comparison to the aircraft flight track data isn't a sufficient validation for the 3 months, in contrast

to the statement on p. 8, line14. Is the model genuinely capturing the boundary layer smoke in July and August? Can the authors create a figure from their model simulation that is comparable to the LASIC data? The authors also refer to several other papers as a form of validation (e.g., p. 2, line28): Deaconu 2019 and Wilcox 2010 rely on satellite datasets that have difficulty distinguishing BB within the BL from sea-spray, Gordon 2018 is a modeling study focusing on a 10-day August time period only that produced an unrealistic 8K warming in the free troposphere, and Ackerman 2004 is for a different location. These references ignore the new information we have thanks to LASIC, ORACLES, and CLARIFY. Besides the studies mentioned by Rev 1, there are also more detailed StCU-to-Cu transition papers indicating the BB can also have a radiative impact in the BL.

Similarly, Kacarab et al. 2020 is also relevant, indicating a kappa of 0.4 for smoke based on oracles observations. This study should be referenced and discussed somewhere, as it does not support the low hygroscopicity for smoke assumed here. https://acp.copernicus.org/articles/20/3029/2020/.

We thank the editor for the comment. The editor's main concern is whether our conclusion that, on the annual average (2016-2017), BB are not the dominant source of $CCN_{0.2\%}$ within the SEA boundary layer is correct? Whether the model significantly underestimates $CCN_{0.2\%}$ from BB within the boundary layer?

To answer the editor's questions. We compared $CCN_{0.2\%}$ simulated by the model (baseline run) to that measured from the LASIC campaign.

[Figure]

Figure 2. Comparison of modelled and observed daily mean $CCN_{0.2\%}$ concentrations. The measured $CCN_{0.2\%}$ is from the LASIC campaign. The modelled $CCN_{0.2\%}$ is from the baseline simulation and interpolated to LASIC coordinates.

The modelled $CCN_{0.2\%}$ concentrations are collocated with observations from LASIC at 340 m on Ascension Island, representing the marine boundary layer CCN in the SEA. Due to the temporal

resolution of the model output, we compared the daily average values. From the figure, the modelled $CCN_{0.2\%}$ is in good agreement with the observation, especially can capture the daily variation of $CCN_{0.2\%}$ during the BB season. The campaign averaged $CCN_{0.2\%}$ is 225 cm$^{-3}$, and the modelled corresponding mean is 239 cm$^{-3}$, with the mean relative error of the modelled $CCN_{0.2\%}$ 6.3%. However, the observed CCN peaks during the BB season are higher than simulations admittedly, indicating that the model is still inadequate for capturing those peak values. Given that we mainly investigate the annual mean CCN in this paper, the small error and the well-matched temporal variability with observation suggest that the model is reasonably in reproducing the CCN in the marine boundary layer in SEA.

We have now included this evaluation in the manuscript. There are several revisions in the manuscript, as shown below.

In the introduction (P4L13-16):

In addition, a ground-based in-situ field measurement campaign (LASIC, Layered Atlantic Smoke Interactions with Clouds) was carried out on Ascension Island, which provided 18 months of observations for aerosols and clouds within the marine boundary layer from June 2016 to October 2017 (Zuidema et al., 2018b).

In the method section (P9-P10)

**2.3 Model evaluation**

The model has been evaluated with the ORACLES (2016, 2017) and CLARIFY measurements by examining the collocated aerosol extinction in our previous paper. The result shows the model can generally capture the spatial and vertical distributions of BB plume (Che et al., 2021). However, as these aircraft observations are mainly located in the free troposphere, we further evaluated modelled CCN within the marine boundary layer using LASIC observations.

[Figure]

Figure 3. Comparison of modelled and observed daily mean $CCN_{0.2\%}$ (CCN at 0.2 % supersaturation) concentrations. The measured $CCN_{0.2\%}$ is from the LASIC campaign. The modelled $CCN_{0.2\%}$ is from the baseline simulation and interpolated to the LASIC coordinates.

The LASIC campaign was carried out on the Atmospheric Radiation Measurement (ARM) Mobile Facility 1 site at Ascension Island, located at a latitude of -7.97°, longitude of-14.35° and altitude of 340.7664 m. The LASIC CCN was measured by a cloud condensation nuclei counter (CCNC-200), which provides the CCN concentration at fixed supersaturations (Roberts and Nenes, 2005; Atmospheric Radiation Measurement (ARM) user facility, 2016). A more detailed description of the sampling location and instruments can be found in the campaign report (Zuidema et al., 2018a). The modelled CCN concentration at 0.2% supersaturation ($CCN_{0.2\%}$) from the baseline simulation is collocated with observations. Due to the temporal resolution of the model output, we compared the daily averages as illustrated in Fig. 3.

As evident in Fig. 3, the modelled $CCN_{0.2\%}$ is in good agreement with the observation, and can capture the daily variation of $CCN_{0.2\%}$ during the BB season. The campaign averaged $CCN_{0.2\%}$ is 225 $cm^{-3}$, and the modelled corresponding means of 239 $cm^{-3}$, with the mean relative error of the modelled $CCN_{0.2\%}$ around 6.3%. However, the observed CCN peaks during the BB season are higher than simulations, indicating that the model is still inadequate for capturing those peak values. One possible reason is that when BC particles have a thick coating, the calculated overall $\kappa$ may be underestimated by the volume mixing rule, which may further underestimate the CCN concentration associated with BB (Kacarab et al., 2020). In addition, uncertainties in the BB emissions, including the magnitude, size and, height of fires, can lead to incorrect estimates of BB aerosol peak concentrations, which can lead to such underestimations of CCN. Given that we mainly investigate the annual mean CCN in this study, the relatively small error and the well-matched temporal variability with observation suggest that the model is fairly reasonably in reproducing the CCN in the marine boundary layer in the SEA. Therefore, this result provides confidence in this study.

In Discussion and conclusion (P20 L3-5)

The model has been evaluated with aircraft measurements from CLARIFY and ORACLES for the aerosol distribution, and is further evaluated in this study with LASIC in-situ observations for the marine boundary layer CCN.

P21 L20-24

By comparing the modelled $CCN_{0.2\%}$ with observations, we find that although the model is generally in good agreement with the measurements, it still underestimates the peak $CCN_{0.2\%}$ during the BB season, suggesting that BB associated $CCN_{0.2\%}$ may be underestimated during the BB season. Also, when BC particles have a thick coating, the calculated $\kappa$ may be underestimated by the volume mixing rule, which may further underestimate CCN concentrations associated with BB.

The modelled annual mean $CCN_{0.2\%}$ in the SEA marine boundary layer during 2016-2017 is 290 $cm^{-3}$, while the mean $CCN_{0.2\%}$ during BB season (July-September) is 331 $cm^{-3}$, 14 % higher than

the annual mean, confirming the influence of BB on $CCN_{0.2\%}$. This is also consistent with the LASIC observation data. For the annual average, BB contributes 17% of the CCN within the marine boundary layer in the SEA, which is lower than anthropogenic sources (26%). Therefore, in terms of emission sources, anthropogenic sources are considered to be the largest source of CCN in this area. However, during the BB season, BB contributed 19% of $CCN_{0.2\%}$, slightly less than anthropogenic sources (21%), indicating the equally important roles of BB and anthropogenic sources to $CCN_{0.2\%}$ during the BB season in this region. Figure 5 in the manuscript and figure S4 in the supplement show the contribution of different sources to the annual mean and BB seasonal mean $CCN_{0.2\%}$, respectively, with the absolute concentrations and relative fractions of $CCN_{0.2\%}$ from each source labelled in the figure. We therefore added the total concentration of annual and BB seasonal averaged $CCN_{0.2\%}$ in their figure caption, respectively.

P12L0-11 (manuscript):

Using the simulation of the present day as the baseline (annual mean $CCN_{0.2\%}$ around 290 cm$^{-3}$), the contribution of each source to $CCN_{0.2\%}$ is marked at the top of the corresponding bar in percentage.

P5L6-8 (supplement)

Using the simulation of the present day as the baseline (BB seasonal mean $CCN_{0.2\%}$ around 331 cm$^{-3}$), the contribution of each source to $CCN_{0.2\%}$ is marked at the top of the corresponding bar in percentage.

We also updated the reference as suggested by the editor. The introduction is revised as follows.

P2 L28- P3 L3

Previous studies suggest that as the BB aerosols are mainly located above and near the inversion layer, when above the inversion layer, the main role of their radiative effect in the SEA is to strengthen the capping inversion and reduce dry air entrainment from cloud tops, thereby increasing the LWP and low-level cloud fraction, resulting in a significant impact on the radiation balance (Wilcox, 2010; Gordon et al., 2018; Deaconu et al., 2019; Mallet et al., 2020; Herbert et al., 2020; Chaboureau et al., 2022). When BB aerosols are located in the marine boundary layer, their radiative effect can enhance the decoupled boundary layer and result in a reduction in cloud cover and LWP, shifting the stratocumulus-to-cumulus transition to the upwind area (Zhang and Zuidema, 2019; Ajoku et al., 2021).

Similarly, Kacarab et al. 2020 is also relevant, indicating a kappa of 0.4 for smoke based on oracles observations. This study should be referenced and discussed somewhere, as it does not support the low hygroscopicity for smoke assumed here.

In the UKESM1 model, $\kappa$ is calculated with the simple volume mixing rule (Petters and Kreidenweis, 2007), which assumes for certain soluble mode aerosols, all particles are homogeneously mixed, so that the overall $\kappa$ is determined based on the volume fraction of the different species and the hygroscopicity of each component. However, since $\kappa$ is set to 0 for BC,

this volumetric mixing rule may underestimate the overall κ of when the BC has a thicker coating. This was illustrated by Kacarab et al. (2020), who found a high overall κ of about 0.4 from eight ORACLES 2017 aircraft observations. However, Zhang et al. (2022) found the overall κ around 0.24 in the marine boundary layer from ORACLES 2018 observations, which may be consistent with our assumption that BB reduces the overall κ.

Therefore, we revised the manuscript as follows.

P5 L13-18 (Method)

The internal volume mixing rule (Petters and Kreidenweis, 2007) is used to calculate the mean hygroscopicity of each mode. Therefore, a higher fraction of less hygroscopic components (e.g. organic and black carbon) can reduce the overall *κ*. However, the overall κ may be underestimated when BC has a thicker coating. This was illustrated by Kacarab et al. (2020), who found a high averaged κ of ~ 0.4 from eight ORACLES 2017 aircraft observations. However, Zhang et al. (2022) found an averaged κ of ~ 0.24 in the marine boundary layer from ORACLES 2018 observations, which is consistent with our assumption that BB reduces the overall κ.

P10L1-4 (Method)

However, the observed CCN peaks during the BB season are higher than simulations, indicating that the model is still inadequate for capturing those peak values. One possible reason is that when BC particles have a thick coating, the calculated overall κ may be underestimated by the volume mixing rule, which may further underestimate the CCN concentration associated with BB (Kacarab et al., 2020).

P21L23-24 (Discussion and conclusion)

Also, when BC particles have a thick coating, the calculated κ may be underestimated by the volume mixing rule, which may further underestimate CCN concentrations associated with BB

The manuscript should also be read over again by a native English speaker, to clarify some of the language. I mention a few specific comments below:

Thanks for the comment. We have carefully proofread the manuscript and have extensively revised it. All revisions are highlighted in the manuscript.

Line 12 p 3: of ->from

Thanks to the editor, we have corrected this.

Line 13 p 3: to the -> to that of Line 16: after -> by

Thanks to the editor, we have corrected these.

p. 3 line 20-21: Kalahari dust doesn't advect far according to https://acp.copernicus.org/articles/21/8169/2021/acp-21-8169-2021.pdf, and it certainly wasn't one of the most observed aerosol at Ascension Island during the LASIC/CLARIFY campaigns. The Begue result for the Netherlands isn't relevant here. The first author could draw on their own work assessing kappa using LASIC measurements. Overall this paragraph and its emphasis on dust lacks support and is misleading.

Thanks for the comment. We have deleted those descriptions.

p. 5: are the BC aerosols internally mixed? Results don't acknowledge that a kappa of 0 for BC doesn't reflect that all of it is likely internally mixed (e.g., Dang et al., 2021), with the BC particle size helping cloud nucleation. How well does the Petters and Kreidenweis internal mixing rule work for this region based on what we know so far from the observations?

Yes. BC aerosols are internally mixed in soluble modes in GLOMAP. This assumes all particles are homogeneously mixed, and their overall κ is based on the volume fraction of different species. However, as the κ of BC is set to 0, this volume mixing calculation can underestimate the overall κ of the aerosol when the BC has a thicker coating, which may further underestimate the CCN concentration. Therefore, we acknowledged the CCN from biomass burning might be underestimated by the model in the discussion, as follows.

P5 L14-19 (Method)

The internal volume mixing rule (Petters and Kreidenweis, 2007) is used to calculate the mean hygroscopicity of each mode. Therefore, a higher fraction of less hygroscopic components (e.g. organic and black carbon) can reduce the overall κ. However, the overall κ may be underestimated when BC has a thicker coating. This was illustrated by Kacarab et al. (2020), who found a high averaged κ of ~ 0.4 from eight ORACLES 2017 aircraft observations. However, Zhang et al. (2022) found an averaged κ of ~ 0.24 in the marine boundary layer from ORACLES 2018 observations, which is consistent with our assumption that BB reduces the overall κ.

P10L1-4 (Method)

However, the observed CCN peaks during the BB season are higher than simulations, indicating that the model is still inadequate for capturing those peak values. One possible reason is that when BC particles have a thick coating, the calculated overall κ may be underestimated by the volume mixing rule, which may further underestimate the CCN concentration associated with BB (Kacarab et al., 2020).

P21L23-24 (Discussion and conclusion)

Also, when BC particles have a thick coating, the calculated κ may be underestimated by the volume mixing rule, which may further underestimate CCN concentrations associated with BB.

P. 5: clarify that anthropogenic does not include BB. You could consider calling it 'non-BB anthropogenic'.

We clarified this in the manuscript as follows.

P5L30-P6L2

Note that although black carbon (BC) and organic carbon (OC) are the main components of BB emissions, these two types of aerosols are also present in anthropogenic emissions. However, the 'anthropogenic' emissions defined here do not include BB aerosols, although BB in southern Africa is associated with human activities (Roberts et al., 2009). In our model setup, BC and OC from our 'anthropogenic' emissions are mainly from fossil fuels and biofuels, and their emission sectors are energy, industrial, shipping, transportation, solvents, waste, agriculture, and residential. In comparison, BC and OC from BB are mainly emitted from the burning of agricultural land, peat, savanna, forest, and deforestation.

Language on nucleation confusing - authors use the same term for gas to particle production of aerosols, and for cloud activation. Here it might be worth adding additional detail to the naming, meaning, to use the longer term of 'aerosol nucleation'. Mention how boundary nucleation differs from total nucleation in the boundary layer for caption of fig. 3. Also, the comment from Rev 2 that the Hadley center models lend more emphasis on nucleation driven CDNC than other models should be mentioned, including the citation to Bellouin et al 2013.

Thanks for the suggestion. We changed the term to 'aerosol nucleation' in the manuscript. The difference between boundary layer nucleation and total nucleation is briefly described in the caption of Fig. 4. (We added a new figure before, thus Fig. 3 in the last version becomes Fig. 4)

P10 L12 – P11 L3

Figure 4. UKESM1 simulated annual mean vertical profiles of CCN concentration at 0.2% supersaturation ($CCN_{0.2\%}$) from different sources (at the standard temperature and pressure STP). Profiles are averaged along the latitudes of the cloud box. The contributions of different sources to $CCN_{0.2\%}$ are listed in (a) to (h), where the contribution of emissions is shown in the yellow frame, and the contribution of atmospheric processes is shown in the light blue frame. Note boundary layer aerosol nucleation is based on organic-mediated aerosol nucleation and is limited to the boundary layer. Total aerosol nucleation includes boundary layer nucleation and homogeneous binary aerosol nucleation in the free troposphere and stratospheric. The contour lines in each subplot are the cloud specific water content from the baseline simulation at the same temporal and spatial average. The same colourmap scale is used in each subplot to facilitate comparison, but the range differs for each plot, corresponding to the maximum and minimum of $CCN_{0.2\%}$.

We also revised the manuscript to address the comment by reviwer#2 about the nucleation.

P16 L4-9

Previous studies have found that more than half of the CCN in the global marine boundary layer is contributed by aerosol nucleation (Clarke et al., 2013; Merikanto et al., 2009b; Williamson et

al., 2019; Clarke and Kapustin, 2002), consistent with our result. However, source attribution in multiple models is recommended to confirm the importance of aerosol nucleation to the CDNC, as the nucleation and Aitken mode aerosol concentrations are significantly overpredicted by HadGEM models (Ranjithkumar et al., 2021; Gordon et al., 2020; Hardacre et al., 2021; Bellouin et al., 2013), suggesting the nucleation contributed CDNC may also be overestimated in our model.

p. 7 line 11: Kacarab et al. 2019 is not consistent with the low model hygroscopicity. p. 8: Note Redemann 2021 shows satellite-derived Nd that are clearly elevated

We have revised the manuscript as follows.

P7L14-19

BB aerosols contributes around 76 % of total AOD in the cloud box during BB season, and can result in a clearly elevated CDNC in the SEA from satellite observations (Redemann et al., 2021), implying the potentially dominant role of BB aerosol in affecting CCN and cloud that motivated the ORACLES, CLARIFY and LASIC campaigns. However, as most of the BB aerosol is above the stratocumulus cloud deck (Fig. 2), combined with a large fraction of low hygroscopic particles such as BC and OC, the fraction of BB aerosol to activate as cloud droplets is uncertain.

We also revised the manuscript about CDNC source attribution as follows.

P16 L20-24

BB aerosols not only can provide CCN to increase CDNC, but also increase CDNC by influencing the vertical distribution of temperature through shortwave absorption, which in turn increases the maximum supersaturation in clouds (Che et al., 2021). This is also evidenced by Fig. 6. As a result, BB becomes the most important emission source of CDNC during the BB season. This result is also supported by a satellite study that found an elevated CDNC with the presence of BB aerosols in this region (Redemann et al., 2021).

p. 16, line 7: to say that 'nucleation is important to CDNC' is very unclear - if you're talking about cloud nucleation, it's overstating the obvious. You don't mean cloud nucleation I recognize, but this is simply not clear writing. Explain what nucleation means in these sentences.

We have revised this as the following.

P16 L9-11

Even during the BB season, the concentration of CDNC contributed by total aerosol nucleation is similar to that of BB (Fig. S5), indicating that total aerosol nucleation remains the most significant source of CDNC throughout the years.

More detail on the non-BB anthro emissions would also be useful. I recognize these are difficult to validate - do the sulfate contributions from the non-BB anthro+sea spray match what was measured at Ascension Island?

We didn't output aerosol chemical compositions from the model, so we cannot directly compare the modelled sulphate with the LASIC campaign.

To validate whether the anthropogenic sources in this region are overestimated, we compared the AOD from anthropogenic sources with the AOD level 1 data observed from AERONET on Ascension Island. This is shown in figure 3.

[Figure]

Figure 3. Monthly mean modelled AOD and box-whisker plots of monthly AOD percentiles (10%, 25%, 50%, 75%, and 90%) measured from AERONET (level 1 data) on Ascension Island. The dashed lines are the AOD from the baseline simulation, and the solid lines are the AOD from the anthropogenic source. Blue and orange colours indicate the AOD at 440 and 670 nm, respectively.

Due to the time setting of model inputs, we compared the monthly average AOD. Note that modelled AOD are averaged over the whole month, while the observations are not available for the whole time during each month, which will result in some uncertainties in the comparison.

The baseline simulated AOD correlates well with the AERONET and has a consistent trend with observations, both being higher in the winter months (also the BB season) for the Southern Hemisphere. In contrast, the modelled values are higher than the AERONET observations during biomass burning season, which may be due to the high proportion of clouds resulting in missing values during these months. Except for January and February of 2017, AOD from anthropogenic sources had a consistent trend with observations. Comparing the values, it can be found that the AOD contributed by anthropogenic sources are smaller than those measured from AERONET, and this difference becomes larger during BB season. Therefore, anthropogenic emissions are likely not overestimated in the model, suggesting the finding in the manuscript that anthropogenic emissions are the largest source of $CCN_{0.2\%}$ in the SEA marine boundary layer is not a result of the model's overestimation of anthropogenic sources. However, as the editor suggests, a comparison of aerosol chemical composition would be more appropriate. However, this is beyond the scope of this work and we hope to discuss this issue in the future.

A bit more effort could be made in the last section mentioning how the new observations can be used to assess and/or improve the model, including using the enhanced resolution of the seasonal cycle.

Thanks for the comment. We revised the manuscript as follows.

[revised manuscript text omitted]